# Towards the Resistance of Neural Network Fingerprinting to Fine-tuning

**Ling Tang**
Shanghai Jiao Tong University
`tling@sjtu.edu.cn`

**Yuefeng Chen**
Alibaba Group
`yuefeng.chenyf@alibaba-inc.com`

**Hui Xue**
Alibaba Group
`hui.xueh@alibaba-inc.com`

**Quanshi Zhang**[*]
Shanghai Jiao Tong University
`zqs1022@sjtu.edu.cn`

## Abstract

This paper proves a new fingerprinting method to embed the ownership information into a deep neural network (DNN) with theoretically guaranteed robustness to fine-tuning. Specifically, we prove that when the input feature of a convolutional layer only contains low-frequency components, specific frequency components of the convolutional filter will not be changed by gradient descent during the fine-tuning process, where we propose a revised Fourier transform to extract frequency components from the convolutional filter. Additionally, we also prove that these frequency components are equivariant to weight scaling and weight permutations. In this way, we design a fingerprint module to embed the fingerprint information into specific frequency components of convolutional filters. Preliminary experiments demonstrate the effectiveness of our method. The source code has been released at https://github.com/tling2000/watermark.

## 1 Introduction

Fingerprinting techniques have long been used to protect the copyright of digital content, including images, videos, and audio [Nematollahi et al., 2017]. Recently, these techniques have been extended to protect the intellectual property of neural networks. Fingerprinting a neural network is usually conducted to implicitly embed the ownership information into the neural network. In this way, if a neural network is stolen or further optimized, the ownership information embedded in the network can be used to verify its true origin. Previous studies usually embedded the ownership information in different ways. For example, Zeng et al. [2024] directly embedded the fingerprint into the network parameters. Kim et al. [2023] used the classification results on a particular type of adversarial examples as the backdoor fingerprint. Kirchenbauer et al. [2023] added a soft fingerprint to the generation result.

However, one of the core challenges of neural network fingerprinting is the theoretically guaranteed resistance to fine-tuning. When network parameters are changed during the fine-tuning process, the fingerprint implicitly embedded in the parameters may be overwritten. Although many studies [Uchida et al., 2017, Adi et al., 2018, Liu et al., 2021, Bansal et al., 2022, Kim et al., 2023, Yang and Wu, 2024, Zeng et al., 2023, Zhang et al., 2024] have realized this problem and have tested the resistance of their fingerprints to fine-tuning, or boosted the resistance in an engineering manner,

---

[*]Quanshi Zhang is the corresponding author <zqs1022@sjtu.edu.cn>. He is with the School of Computer Science, at the Shanghai Jiao Tong University, China.

39th Conference on Neural Information Processing Systems (NeurIPS 2025).

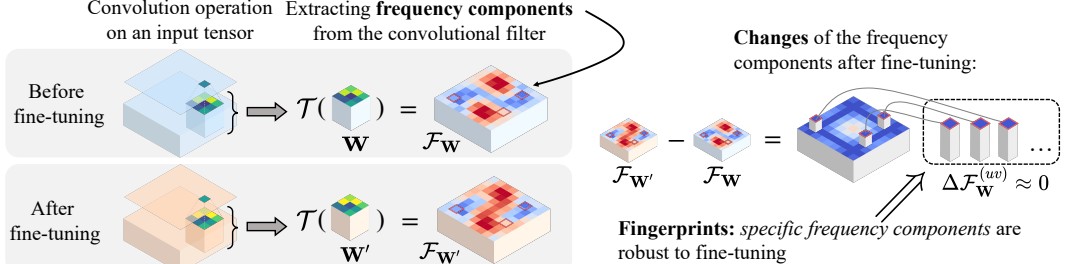

Figure 1: Overview of our theory. We prove that the specific frequency components[3] $\mathcal{F}_{\mathbf{W}}^{(uv)}$, which are obtained by conducting a revised discrete Fourier transform $\mathcal{T}(\cdot)$ on the convolutional filter $\mathbf{W}$, keep stable in the training process. Thus, these specific frequency components $\mathcal{F}_{\mathbf{W}}^{(uv)}$ are used as robust fingerprints to fine-tuning.

there is no theoretically designed metric that is intrinsically resistant to fine-tuning in mathematics, to the best of our knowledge.

To this end, the core challenge towards the resistance to fine-tuning is to explore an invariant term in the neural network to fine-tuning, *e.g.*, certain network parameters or some properties of network parameters that are least affected during the fine-tuning process. Although Zeng et al. [2023] and Zhang et al. [2024] have explored invariant terms *w.r.t.* weight scaling and weight permutations for fingerprinting, the theoretically guaranteed invariant term to fine-tuning remains unsolved.

Therefore, in this study, we aim to discover and prove such an invariant term to fine-tuning. Specifically, as Figure 2 shows, Tang et al. [2023] have found that the forward propagation through a convolutional layer $\mathbf{W} \otimes \mathbf{X} + b \cdot \mathbf{1}_{M \times N}$ can be reformulated as a specific vector multiplication between frequency components $\mathcal{F}_{\mathbf{W}}^{(uv)} \cdot \mathcal{F}_{\mathbf{X}}^{(uv)} + \delta_{uv} MNb$ in the frequency domain, where $\mathcal{F}_{\mathbf{X}}^{(uv)}$ denotes the frequency component of the input feature $\mathbf{X}$ at frequency $(u, v)$, which is extracted by conducting a discrete Fourier transform, $\mathcal{F}_{\mathbf{W}}^{(uv)}$ denotes the frequency component[2] of the convolutional filter $\mathbf{W}$, and $b$ is the bias term.

Based on this, we prove that if the input feature $\mathbf{X}$ only contains the low-frequency components, *then specific frequency components of a convolutional filter $\mathcal{F}_{\mathbf{W}}^{(uv)}$ are stable w.r.t. network fine-tuning*. Additionally, we also prove that these specific frequency components[2] exhibit equivariance to weight scaling and weight permutations.

Therefore, as Figure 1 shows, we propose to use such frequency components[3] $\mathcal{F}_{\mathbf{W}}^{(uv)}$ as the robust fingerprint. Besides, the overwriting attack is another important issue for fingerprinting. To defend the fingerprint from the overwriting attack, we introduce an additional loss to train the model, which ensures that the overwriting of the fingerprint will significantly hurt the model's performance.

The contribution of this study can be summarized as follows. (1) We discover and theoretically prove that specific frequency components of a convolutional filter remain invariant during training and are equivariant to weight scaling and weight permutations. (2) Based on the theory, we propose to encode the fingerprint information into these frequency components, so as to ensure that the fingerprint is robust to fine-tuning, weight scaling, and weight permutations. (3) Preliminary experiments have demonstrated the effectiveness of the proposed method.

## 2    Related Work

The robustness of fingerprints has always been a key issue in the field of neural network fingerprinting. In this paper, we limit our discussion to the fingerprint embedded in network parameters for the

---

[2] The frequency component $\mathcal{F}_{\mathbf{W}}^{(uv)}$ of the convolutional filter is defined in Equation (4), which is extracted by applying a revised discrete Fourier transform on the convolutional filter $\mathbf{W}$. According to Theorem 3.1, the frequency component $\mathcal{F}_{\mathbf{W}}^{(uv)}$ at frequency $(u, v)$ represents the influence of the convolutional filter $\mathbf{W}$ on the corresponding frequency component $\mathcal{F}_{\mathbf{X}}^{(uv)}$ extracted from the input feature $\mathbf{X}$.

[3] For clarity, we move low frequencies to the center of the spectrum map, and move high frequencies to the corners of the spectrum map.

Table 1: Methods for model source tracing. "✓" indicates robustness, "✗" indicates lack of robustness, "–" means that the attack is no applicable, and "NTD" means that there is no target design to defend against the attack.

| Method | Robustness against attacks | | | | | |
| | Fine-tuning | Permutation | Scaling | Overwriting | Pruning | Distillation |
|---|---|---|---|---|---|---|
| [Adi et al., 2018] | NTD | ✓ | ✓ | – | No test | NTD |
| [Jia et al., 2021] | NTD | ✓ | ✓ | – | Tested | ✓ |
| [Bansal et al., 2022] | Enhanced via noise training | ✓ | ✓ | – | No test | ✓ |
| [Kim et al., 2023] | Enhanced via trigger confidence boosting | ✓ | ✓ | – | Tested | ✓ |
| [Gubri et al., 2024] | NTD | ✓ | ✓ | – | No test | NTD |
| [Szyller et al., 2021] | – | – | – | – | – | ✓ |
| [Charette et al., 2022] | – | – | – | – | – | ✓ |
| [Fan et al., 2019] | Enhanced via fingerprint amplification | – | – | ✓ | Tested | ✗ |
| [Zhang et al., 2020] | Enhanced via fingerprint amplification | – | – | ✓ | Tested | ✗ |
| [Uchida et al., 2017] | Empirical, NTD | ✗ | ✗ | ✗ | Tested | ✗ |
| [Yang and Wu, 2024] | Empirical, NTD | ✓ | ✓ | – | No test | ✗ |
| [Zeng et al., 2024] | Empirical, NTD | ✓ | ✓ | ✗ | No test | ✗ |
| [Zhang et al., 2024] | Empirical, NTD | ✓ | ✓ | – | Tested | ✗ |
| **Ours** | **Theoretically guaranteed** | ✓ | ✓ | ✓ | No test | ✗ |

protection of the DNN's ownership information. Especially, we discuss previous methods under attacks of fine-tuning, weight scaling, weight permutations, pruning, and distillation.

Weight scaling and weight permutations are typical attacking methods which change the fingerprint by rearranging the network's parameters. Therefore, Zeng et al. [2023] embedded the fingerprint information into the multiplication of specific weight matrices, which were invariant to weight scaling and weight permutations. Zhang et al. [2024] measured the CKA similarity Kornblith et al. [2019] between the features of different layers in a DNN as the robust fingerprint towards weight scaling and weight permutations.

Compared to the robustness to weight scaling and weight permutations, the robustness to fine-tuning presents a more significant challenge. Up to now, there is no theoretically guaranteed robust fingerprint to fine-tuning, to the best of our knowledge. Thus, many fingerprint techniques were implemented in an engineering manner to defend the fine-tuning against attack. Liu et al. [2021] selected network parameters, which did not change a lot during fine-tuning, to encode the fingerprint information. Kim et al. [2023] used the classification results on a trigger set as the fingerprint and improved robustness by increasing the classification confidence. Zeng et al. [2023] found that the direction of the vector formed by all parameters was relatively stable during fine-tuning, so as to use it as the fingerprint.

However, Chen et al. [2019], Aiken et al. [2021], Shafieinejad et al. [2021] and Xu et al. [2024] showed that, despite various engineering defense methods, most fingerprints could still be effectively removed from the neural network under certain fine-tuning settings.

Therefore, a theoretically certified robust fingerprint is of considerable value in both theory and practice. To this end, Bansal et al. [2022] and Ren et al. [2023] proposed to use the classification results on a trigger set as the fingerprint and proved that the classification accuracy was lower bounded when the attacker did not change the network's parameters by more than a distance in terms of $l_p$-norm ($p > 1$). These methods proved a safe range of parameter changes during fine-tuning, but they did not propose an intrinsically robust fingerprint.

In contrast, we have proved a theoretically guaranteed robust fingerprint to fine-tuning, *i.e.*, proving that the convolutional filter's specific frequency components[2] keep stable during fine-tuning. We summarize representative methods for model source tracing in Table 1.

## 3  Method

### 3.1  Preliminaries: reformulating the convolution in the frequency domain

In this subsection, we reformulate the forward propagation through a convolutional filter in the frequency domain, *i.e.*, Tang et al. [2023] have proven Theorem 3.1, showing that the convolution

operation can be reformulated as the vector multiplication in the frequency domain as shown in Figure 2. Specifically, let us focus on a convolutional filter with $C$ channels and a kernel size of $K \times K$. The convolutional filter is parameterized by weights $\mathbf{W} \in \mathbb{R}^{C \times K \times K}$ and the bias term $b \in \mathbb{R}$. Accordingly, we apply this filter to an input feature $\mathbf{X} \in \mathbb{R}^{C \times M \times N}$, and obtain an output feature map $Y \in \mathbb{R}^{M' \times N'}$.

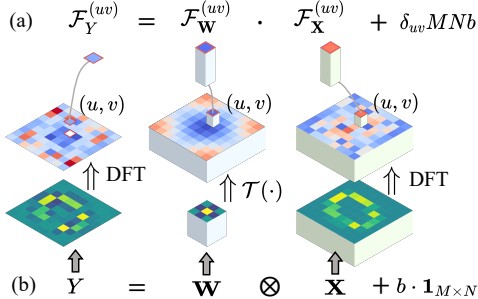

$$Y = \mathbf{W} \otimes \mathbf{X} + b \cdot \mathbf{1}_{M' \times N'}, \qquad (1)$$

where $\otimes$ denotes the convolution operation. $\mathbf{1}_{M' \times N'}$ is an $M' \times N'$ matrix, in which elements are all ones.

Figure 2: The convolution operation in the spatial domain (a) can be reformulated as the vector multiplication in the frequency domain (b)[3].

**Background 1: notation for frequency components of the input feature and the output feature.** Before reformulating the convolution operation, we first introduce the notation for frequency components. Given the input feature $\mathbf{X} \in \mathbb{R}^{C \times M \times N}$ and the output feature $Y \in \mathbb{R}^{M \times N}$, we conduct the two-dimensional DFT on each $c$-th channel $X^{(c)} \in \mathbb{R}^{M \times N}$ of $\mathbf{X}$ and the output matrix $Y$ to obtain the frequency element $G_{uv}^{(c)} \in \mathbb{C}$ and $H_{uv} \in \mathbb{C}$ at frequency $(u, v)$ as follows. $\mathbb{C}$ denotes the set of complex numbers.

$$G_{uv}^{(c)} = \sum_{m=0}^{M-1} \sum_{n=0}^{N-1} X_{mn}^{(c)} e^{-i(\frac{um}{M} + \frac{vn}{N})2\pi}, \qquad H_{uv} = \sum_{m=0}^{M-1} \sum_{n=0}^{N-1} Y_{mn} e^{-i(\frac{um}{M} + \frac{vn}{N})2\pi}, \qquad (2)$$

where $X_{mn}^{(c)}, Y_{mn} \in \mathbb{R}$ denote the elements of $X^{(c)}$ and $Y$ at position $(m, n)$, respectively.

For clarity, we can organize all frequency elements belonging to the same $c$-th channel to construct a frequency spectrum matrix $G^{(c)} \in \mathbb{C}^{M \times N}$. Alternatively, we can also re-organize these frequency elements at the same frequency $(u, v)$ to form a frequency component vector $\mathcal{F}_{\mathbf{X}}^{(uv)} \in \mathbb{C}^C$.

$$\forall u, v, \ \mathcal{F}_{\mathbf{X}}^{(uv)} = \left[ G_{uv}^{(1)}, G_{uv}^{(2)}, \ldots, G_{uv}^{(C)} \right]^\top \in \mathbb{C}^C, \qquad \forall c, \ G^{(c)} = \begin{bmatrix} G_{00}^{(c)} & \cdots \\ \vdots & \ddots \end{bmatrix} \in \mathbb{C}^{M \times N}. \qquad (3)$$

Similarly, $\mathcal{F}_Y^{(uv)} = H_{uv} \in \mathbb{C}$ represents the frequency component of the output feature $Y$ at the frequency $(u, v)$.

**Background 2: notation for frequency components of the convolutional filter.** By following [Tang et al., 2023], the frequency component[2] $\mathcal{F}_{\mathbf{W}}^{(uv)}$ of the convolutional filter $\mathbf{W} \in \mathbb{R}^{C \times K \times K}$ at frequency $(u, v)$ is defined as follows, which is computed by conducting the revised discrete Fourier transform $\mathcal{T}_{uv}(\cdot)$ of frequency $(u, v)$ on $\mathbf{W}$.

$$\forall u, v, \ \mathcal{F}_{\mathbf{W}}^{(uv)} = \mathcal{T}_{uv}(\mathbf{W}) = [Q_{uv}^{(1)}, Q_{uv}^{(2)}, \ldots, Q_{uv}^{(C)}]^\top \in \mathbb{C}^c, \qquad (4)$$

where $Q_{uv}^{(c)} = \sum_{t=0}^{K-1} \sum_{s=0}^{K-1} W_{ts}^{(c)} e^{i(\frac{ut}{M} + \frac{vs}{N})2\pi}$, $W_{ts}^{(c)}$ denotes the element at position $(t, s)$ of the $c$-th channel $W^{(c)} \in \mathbb{R}^{K \times K}$ of $\mathbf{W}$.

For all frequency components $\mathcal{F}_{\mathbf{X}}^{(uv)} \in \mathbb{C}^C$, $\mathcal{F}_Y^{(uv)} \in \mathbb{C}$ and $\mathcal{F}_{\mathbf{W}}^{(uv)} \in \mathbb{C}^C$, frequency $(u, v)$ close to $(0, 0), (0, N-1), (M-1, 0)$, or $(M-1, N-1)$ represents the low frequency, while frequency $(u, v)$ close to $(M/2, N/2)$ is considered as the high frequency.

For notational convenience, we can use tensors $\mathcal{F}_{\mathbf{X}} \in \mathbb{C}^{C \times M \times N}$ and $\mathcal{F}_{\mathbf{W}} = \mathcal{T}(\mathbf{W}) \in \mathbb{C}^{C \times M \times N}$ to denote the tensors of all frequency components of $\mathbf{X}$ and $\mathbf{W}$.

$$\mathcal{F}_{\mathbf{X}} = \begin{bmatrix} \mathcal{F}_{\mathbf{X}}^{(00)} & \cdots \\ \vdots & \ddots \end{bmatrix} \in \mathbb{C}^{C \times M \times N}, \quad \mathcal{F}_{\mathbf{W}} = \mathcal{T}(\mathbf{W}) = \begin{bmatrix} \mathcal{F}_{\mathbf{W}}^{(00)} & \cdots \\ \vdots & \ddots \end{bmatrix} \in \mathbb{C}^{C \times M \times N}. \qquad (5)$$

**Reformulating the convolution operation in the frequency domain.** Based on the above notation, Tang et al. [2023] have proven that the forward propagation of the convolution operation in Equation (1) can be reformulated as a vector multiplication in the frequency domain as follows.

**Theorem 3.1.**

$$\begin{aligned} Y = \mathbf{W} \otimes \mathbf{X} + b \cdot \mathbf{1}_{M \times N} \quad &\Longleftrightarrow \quad \mathcal{F}_Y^{(uv)} = \mathcal{F}_{\mathbf{W}}^{(uv)} \cdot \mathcal{F}_{\mathbf{X}}^{(uv)} + \delta_{uv} MNb, \\ (\textit{Spatial domain}) \quad & \qquad\qquad (\textit{Frequency domain}) \end{aligned} \qquad (6)$$

*where · denotes the scalar product of two vectors; $\delta_{uv}$ is defined as $\delta_{uv} = 1$ if and only if $u = v = 0$, and $\delta_{uv} = 0$ otherwise. In particular, the convolution operation $\otimes$ is conducted with circular padding Jain [1989] and a stride size of 1, which avoids changing the size of the output feature (i.e., ensuring $M' = M$ and $N' = N$).*

## 3.2 Invariant frequency components of the convolutional filter

In this subsection, we aim to prove that frequency components of the convolutional filter $\mathcal{F}_{\mathbf{W}}^{(uv)}$ at certain frequencies $(u, v)$ are relatively stable during training. Additionally, these frequency components are also equivariant to other attacks like weight scaling and weight permutations. In this way, we can embed the fingerprints into these components to enhance their resistance to fine-tuning, weight scaling, and weight permutations.

**Proving specific frequency components of the filter are invariant towards fine-tuning.** Specifically, based on the forward propagation in the frequency domain formulated in Equation (6), we prove that if the input feature $\mathbf{X}$ contains only the fundamental frequency components, *i.e.*, $\forall (u, v) \neq (0, 0), \mathcal{F}_{\mathbf{X}}^{(uv)} = 0$, then frequency components $\mathcal{F}_{\mathbf{W}}^{(uv)}$ at specific frequencies will not change over the training process.

To prove the invariance of the frequency components towards fine-tuning, we decompose the entire training process into massive steps of gradient descent optimization. Each step of gradient descent optimization *w.r.t.* the loss function can be formulated as $\mathbf{W}' = \mathbf{W} - \eta \frac{\partial Loss}{\partial \mathbf{W}}$. Let $\mathcal{F}_{\mathbf{W}}^{(uv)} = \mathcal{T}_{uv}(\mathbf{W})$ denote the frequency component extracted from the filter $\mathbf{W}$ before the optimization, according to Equation (4). Let $\mathcal{F}_{\mathbf{W}'}^{(uv)} = \mathcal{T}_{uv}(\mathbf{W} - \eta \frac{\partial Loss}{\partial \mathbf{W}})$ denote the frequency component extracted from the updated filter $\mathbf{W}'$ after the step of gradient descent optimization.

**Theorem 3.2.** *(**The change of frequency components during training**, proven in Appendix A.1) The change of each frequency component $\mathcal{F}_{\mathbf{W}}^{(uv)}$ before and after a single-step gradient descent optimization is reformulated as follows.*

$$\Delta \mathcal{F}_{\mathbf{W}}^{(uv)} = \mathcal{T}_{uv}(\mathbf{W} - \eta \frac{\partial Loss}{\partial \mathbf{W}}) - \mathcal{T}_{uv}(\mathbf{W}) = \mathcal{F}_{\mathbf{W}'}^{(uv)} - \mathcal{F}_{\mathbf{W}}^{(uv)} = -\eta \sum_{u'=0}^{M-1} \sum_{v'=0}^{N-1} A_{uvu'v'} \frac{\partial Loss}{\partial \overline{\mathcal{F}}_{Y}^{(u'v')}} \cdot \overline{\mathcal{F}}_{\mathbf{X}}^{(u'v')},$$
(7)

*where $A_{uvu'v'} = \frac{\sin(\frac{K(u-u')\pi}{M})}{\sin(\frac{(u-u')\pi}{M})} \frac{\sin(\frac{K(v-v')\pi}{N})}{\sin(\frac{(v-v')\pi}{N})} \cdot e^{i(\frac{(K-1)(u-u')}{M} + \frac{(K-1)(v-v')}{N})\pi} \in \mathbb{C}$ is a complex coefficient; $\overline{\mathcal{F}}_{\mathbf{X}}^{(u'v')}$ denotes the conjugate of $\mathcal{F}_{\mathbf{X}}^{(u'v')}$.*

Corollary 3.3 shows that if the input feature $\mathbf{X}$ only contains the fundamental frequency component, then the specific frequency components $\mathcal{F}_{\mathbf{W}}^{(uv)}$ keep unchanged over the training process. This is an ideal case where the input feature only contains the fundamental frequency component, and $u, v$ can take non-integer values.

**Corollary 3.3.** *(**Invariant frequency components towards fine-tuning**, proven in Appendix A.2) In the training process, if the input feature $\mathbf{X}$ only contains the fundamental frequency component, i.e., $\forall (u, v) \neq (0, 0), \mathcal{F}_{\mathbf{X}}^{(uv)} = 0$, then frequency components $\mathcal{F}_{\mathbf{W}}^{(uv)}$ at all the following frequencies keep invariant.*

$$\Delta \mathcal{F}_{\mathbf{W}}^{(uv)} = \mathbf{0}, \quad s.t. \quad u = \frac{iM}{K} \ or \ v = \frac{jN}{K}, \quad i, j \in \{1, 2, \ldots, K-1\}. \tag{8}$$

However, in real applications, the input feature usually contains low-frequency components, and frequencies must be integers when conducting the DFT. Under these conditions, each element in $\Delta \mathcal{F}_{\mathbf{W}}^{(uv)}$ is nearly zero at integer frequencies that are close to the frequencies specified above.

**Proposition 3.4.** *If the input feature $\mathbf{X}$ only contains the low-frequency components, i.e., $\forall (u, v) \notin S_r^{low}, \mathcal{F}_{\mathbf{X}}^{(uv)} = 0$, s.t. $S_r^{low} = \{(u, v) | u \in [0, r] \cup [M - r, M), v \in [0, r] \cup [N - r, N)\}$, then frequency components $\mathcal{F}_{\mathbf{W}}^{(uv)}$ at all following frequencies keep relative stable during the training process.*

$$\Delta \mathcal{F}_{\mathbf{W}}^{(uv)} \approx \mathbf{0}, \quad s.t. \quad u = \lfloor \frac{iM}{K} \rceil \ or \ v = \lfloor \frac{jN}{K} \rceil, \quad i, j \in \{1, 2, \ldots, K-1\}, \tag{9}$$

*where $\lfloor x \rceil$ is used to round the real number $x$ to the nearest integer; $r$ is a positive integer ($r \leq 2$).*

**Proving frequency components are equivariant to weight scaling.** The scaling attack [Yan et al., 2023] means scaling the weights of a convolutional layer by a constant $a$, and scaling the weights

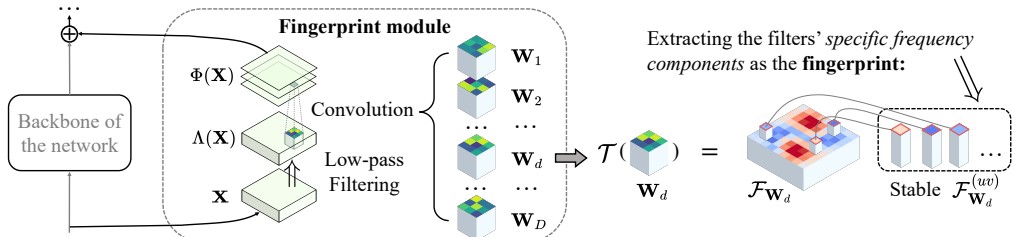

Figure 3: Architecture of the fingerprint module. The fingerprint module is connected in parallel to the backbone of the neural network. We extract the specific frequency components[3] from the convolutional filters in the fingerprint module as the network's fingerprint.

of the next convolutional layer by the inverse proportion $1/a$. In this way, the model's performance will not be affected, but the fingerprint embedded in the weights usually will change. We prove Theorem 3.5, which shows that the frequency components are equivariant to the scaling attack.

**Theorem 3.5.** *(**Equivariance towards weight scaling**, proven in Appendix A.3) If we scale all weights in the convolutional filter $\mathbf{W}$ by a constant $a$ as $\mathbf{W}^* = a \cdot \mathbf{W}(a > 0)$, then the frequency components of $\mathbf{W}^*$ are equal to the scaled frequency components of $\mathbf{W}$, as follows.*

$$\forall u, v, \quad \mathcal{F}_{\mathbf{W}^*}^{(uv)} = a \cdot \mathcal{F}_{\mathbf{W}}^{(uv)}. \tag{10}$$

**Proving frequency components are equivariant to weight permutations.**    The permutation attack [Yan et al., 2023] on the filters means permuting the filters and corresponding bias terms of a convolutional layer, and then permuting the channels of every filter in the next convolutional layer in the same order. Thus, the network's outputs remain unaffected, while the fingerprint embedded in the weights is altered. Theorem 3.6 proves the equivariance of the frequency components when permuting convolutional filters.

**Theorem 3.6.** *(**Equivariance towards weight permutations**, proven in Appendix A.4) Let us consider a convolutional layer with $D$ convolutional filters with $D$ bis terms arranged as $\mathbb{W} = [\mathbf{W}_1, \mathbf{W}_2, \cdots, \mathbf{W}_D]$ and $\mathbf{b} = [b_1, b_2, \cdots, b_D]$. If we use a permutation $\pi$ to rearrange the above filters and bias terms as $\pi\mathbb{W} = [\mathbf{W}_{\pi(1)}, \mathbf{W}_{\pi(2)}, \cdots, \mathbf{W}_{\pi(D)}]$ and $\pi\mathbf{b} = [b_{\pi(1)}, b_{\pi(2)}, \cdots, b_{\pi(D)}]$, where $[\pi(1), \pi(2), \cdots, \pi(D)]$ is a random permutation of integers from $1$ to $D$, then the frequency components of $\pi\mathbb{W}$ are equal to the permuted frequency components of $\mathbb{W}$, as follows.*

$$\forall \pi, u, v, \quad \left[ \mathcal{F}_{\mathbf{W}_{\pi(1)}}^{(uv)}, \cdots, \mathcal{F}_{\mathbf{W}_{\pi(D)}}^{(uv)} \right] = \pi \left[ \mathcal{F}_{\mathbf{W}_1}^{(uv)} \cdots \mathcal{F}_{\mathbf{W}_D}^{(uv)} \right], \tag{11}$$

*where $\mathcal{F}_{\mathbf{W}_d}^{(uv)} = \mathcal{T}_{uv}(\mathbf{W}_d) \in \mathbb{C}^C$ denotes the frequency components extracted from the $d$-th filter $\mathbf{W}_d$ at frequency $(u, v)$.*

### 3.3    Using the invariant frequency components as the neural network's fingerprint

In the last subsection, we prove that if the input feature only contains the low-frequency components, the filter's frequency components $\mathcal{F}_{\mathbf{W}}^{(uv)}$ at specific frequencies $(u, v)$ keep stable during training. Furthermore, these components exhibit equivariance to weight scaling and weight permutations.

**Fingerprint module.**    All the above findings and proofs enable us to use the specific frequency components as the fingerprint of the neural network. In this way, the fingerprint will be highly robust to fine-tuning, weight scaling, and weight permutations. Specifically, as Figure 3 shows, we construct the following fingerprint module $\Phi(\mathbf{X})$ to contain the fingerprint, which consists of a low-pass filter $\Lambda(\cdot)$ and a convolutional layer with $D$ convolutional filters $\mathbb{W} = [\mathbf{W}_1, \mathbf{W}_2, \cdots, \mathbf{W}_D]$ and $D$ bias terms $\mathbf{b} = [b_1, b_2, \cdots, b_D]$.

$$\Phi(\mathbf{X}) = [Y_1, Y_2, \cdots, Y_D], \quad s.t. \quad Y_d = \mathbf{W}_d \otimes \Lambda(\mathbf{X}) + b_d \cdot \mathbf{1}_{M \times N}, \tag{12}$$

where the low-pass filtering operation $\Lambda(\cdot)$ preserves frequency components in $\mathbf{X}$ at low frequencies in $S_r^{\text{low}} = \{(u, v) | u \in [0, r] \cup [M - r, M), v \in [0, r] \cup [N - r, N)\}$ ($r \leq 2$) and removes all other frequency components, *i.e.*, setting $\forall (u, v) \notin S_r^{\text{low}}, \mathcal{F}_{\mathbf{X}}^{(uv)} = 0$. $\mathbf{1}_{M \times N}$ is an $M \times N$ matrix, in which elements are all ones.

**Setting invariant frequency components as the fingerprint.**    *In this way, when we extract frequency components $\mathcal{F}_{\mathbf{W}_d}^{(uv)}$ from each $d$-th convolutional filter $\mathbf{W}_d$ in the fingerprint module $\Phi(\mathbf{X})$ based on*

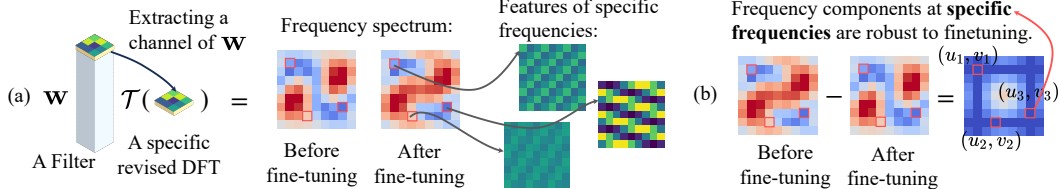

Figure 4: (a) We show the matrix of frequency components[3] extracted from a single channel of the convolutional filter. (b) Specific feature components are robust to fine-tuning.

*Equation (4), we can consider the frequency components at the following frequencies in the set $S'$, as the fingerprint.*

$$\left[\mathcal{F}_{\mathbf{W}_1}^{(uv)}, \mathcal{F}_{\mathbf{W}_2}^{(uv)}, \cdots, \mathcal{F}_{\mathbf{W}_D}^{(uv)}\right] \quad s.t. \quad (u,v) \in S', \tag{13}$$

*where $S' = \{(u,v)|u = \lfloor iM/K \rceil$ or $v = \lfloor jN/K \rceil; i,j \in \{1,2,\ldots,K-1\}\}$. According to Proposition 3.4, the fingerprint will keep stable during training.*

*Implementation details.* We notice that in the fingerprint module, the low-pass filter $\Lambda(\cdot)$ may hurt the flexibility of feature representations. Therefore, as Figure 3 shows, the fingerprint module is connected in parallel to the backbone architecture of the neural network. In this way, this design does not significantly change the network's architecture or seriously hurt its performance. Unless stated otherwise, we set the integer $r = 1$, the kernel size $K = 3$, and the filter number $D = 256$.

**Verifying the invariance towards fine-tuning.** We conducted experiments to verify the invariance of the fingerprint frequency components towards fine-tuning. We computed the average $l_2$-norm of the change of the frequency components $\mathbb{E}_d[\|\Delta\mathcal{F}_{\mathbf{W}_d}^{(uv)}\|]$, where $\Delta\mathcal{F}_{\mathbf{W}_d}^{(uv)}$ denotes the change of the frequency components extracted from the $d$-th convolutional filter after fine-tuning. Please see Appendix B.2 for more details. The experimental results are shown in Figure 5, indicating that the frequency components used as the fingerprint are robust to fine-tuning.

**Visualization of the fingerprint.** Figure 4(a) shows the feature maps when we apply the inverse discrete Fourier transform (IDFT) to each specific unit in the fingerprint feature components. Figure 4(b) shows the specific frequencies in the set $S'$ used as the fingerprint. Notably, the revised transformation of the convolutional filter in Equation (4) is irreversible, so the feature maps can illustrate features of specific frequencies that are affected by the selected frequency components[2] $\mathcal{F}_{\mathbf{W}}^{(uv)}$ in the filter $\mathbf{W}$.

### 3.4 Detecting the fingerprint

In this subsection, we introduce how to detect the fingerprint. We are given a source fingerprinted DNN with a fingerprint module containing $D$ convolutional filters $[\mathbf{W}_1, \mathbf{W}_2, \cdots, \mathbf{W}_D]$ and a suspicious DNN with a fingerprint module also containing $D$ convolutional filters $[\mathbf{W}'_1, \mathbf{W}', \cdots, \mathbf{W}'_D]$. We aim to detect whether the suspicious DNN is obtained from the source DNN by fine-tuning, weight scaling, or weight permutations.

Considering the permutation attack, the detection towards the frequency components should consider the matching between the frequency components of different convolutional filters of the two DNNs, *i.e.*, we can definitely find a permutation $[\pi(1), \pi(2), \cdots, \pi(D)]$ to assign each $d$-th convolutional filter $\mathbf{W}_d$ in the source DNN with the $\pi(d)$-th filter $\mathbf{W}'_{\pi(d)}$ in the suspicious DNN. Specifically, we use the following fingerprint score $\rho \in [0,1]$ between two DNNs to identify the matching quality.

$$\rho = \max_\pi \mathbb{E}_{(u,v)\in S'}\mathbb{E}_d\left[\mathbb{I}(\cos(\mathcal{F}_{\mathbf{W}_d}^{(uv)}, \mathcal{F}_{\mathbf{W}'_{\pi(d)}}^{(uv)}) \geq \tau)\right], \tag{14}$$

where $\mathbb{I}(\cdot)$ denotes an indicator function that returns $1$ if $\cos(\mathcal{F}_{\mathbf{W}_d}^{(uv)}, \mathcal{F}_{\mathbf{W}'_{\pi(d)}}^{(uv)}) \geq \tau$, and returns $0$ otherwise. $\cos(\mathcal{F}_{\mathbf{W}_d}^{(uv)}, \mathcal{F}_{\mathbf{W}'_{\pi(d)}}^{(uv)})$ denotes the cosine similarity[4] of the frequency components. $\tau$ is a threshold, and we set $\tau = 0.99$ in this paper unless otherwise stated. Thus, we can use the fingerprint

---

[4]The cosine similarity between two complex vectors lies in the range $[-1,1]$, where larger values indicate higher similarity. Please see Appendix C.1 for details.

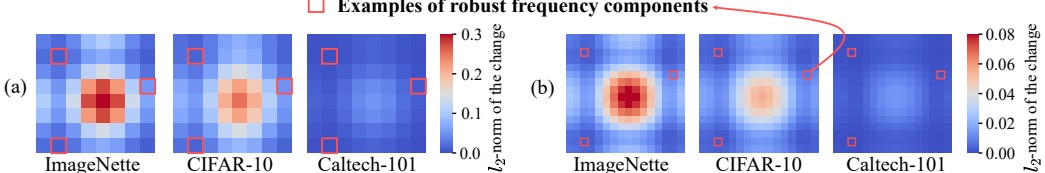

Figure 5: Significance of the change of frequency components[3] after fine-tuning. The red color indicates a high $l_2$-norm of the change of the frequency component at $(u, v)$, given as $\mathbb{E}_d[\|\Delta\mathcal{F}_{\mathbf{W}_d}^{(uv)}\|]$, which is averaged across all filters. (a) The results for AlexNet fine-tuned on the ImageNette, CIFAR-10, and Caltech-101 datasets. (b) The results for ResNet-18 fine-tuned on the same datasets.

score $\rho$ to determine whether the suspicious network originates from the source network by checking $\rho > t$, where the setting of threshold $t$ will be introduced in Section 4.

### 3.5 Learning towards the overwriting attack

Another challenge is the resistance of the fingerprint to the overwriting attack. Given the fingerprint module's architecture, if the attacker has obtained the authority to edit its parameters, then he can overwrite the weights $\mathbb{W}$ in the fingerprint module with entirely new values to change the fingerprint.

Let us consider a DNN for the classification of $n$ categories. To defend the overwriting attack, the basic idea is to construct the $(n + 1)$-th category as a pseudo category besides the existing $n$ categories. If the network is under an overwriting attack, then the neural network is trained to classify all samples into the pseudo category. Thus, the neural network's vulnerability to overwriting attacks can effectively prevent attacker from conducting such attacks to the neural network, *i.e.*, the attacker would not obtain a high-performance network without the fingerprint. Therefore, we train the network by adding an additional loss $\mathcal{L}_{\text{attack}}$, which pushes the attacked network to classify all samples into the pseudo category, to the standard cross-entropy loss $\mathcal{L}_{\text{CE}}$ for multi-category classification.

$$\begin{aligned}
\mathcal{L}(\mathbb{W}, \mathbf{b}, \theta | x) &= \mathcal{L}_{\text{CE}}(\mathbb{W}, \mathbf{b}, \theta | x) + \mathcal{L}_{\text{attack}}(\mathbb{W}, \mathbf{b}, \theta | x) \\
&= -\sum_{k=1}^{n} p(y = k|x)\log q(y = k|x; \mathbb{W}, b, \theta) - \lambda \cdot \log q(y = n + 1|x; \mathbb{W} + \epsilon, b, \theta),
\end{aligned} \tag{15}$$

where $x$ denotes an input sample, and $y$ denotes its corresponding label; $\theta$ denotes the DNN's parameters. $q(y = k|x; \mathbb{W}, \mathbf{b}, \theta)$ denotes the classification probability predicted by the DNN. $p(y = k|x)$ is the ground truth probability. The scalar weight $\lambda$ balances the influence of $\mathcal{L}_{\text{CE}}$ and $\mathcal{L}_{\text{attack}}$.

In the above loss function, we add a random noise[5] $\epsilon$ to the parameters $\mathbb{W}$ in the fingerprint module to mimic the state of the neural network with an overwritten fingerprint. To enhance the module's sensitivity to such attacks, we do not completely overwrite the parameters but add random noise.

*Ablation studies.* We conducted an ablation experiment to evaluate the effectiveness of the newly added loss term $\mathcal{L}_{\text{attack}}$. We compared the classification accuracy of the network without the attack and the classification accuracy under the attack to analyze the performance decline of the network towards the overwriting attack. Please see Appendix B.1 for results and more details. The results indicated that the loss term $\mathcal{L}_{\text{attack}}$ made the fingerprint more resistant to overwriting attacks.

## 4 Experiments

**Verifying the robustness towards the combined attacks.** We conducted experiments to compare the proposed fingerprint with other competing methods for model source tracing under combined attacks, where the permutation attack, the scaling attack, and the fine-tuning attack were applied sequentially. Specifically, let us consider a source DNN, denoted by $\mathcal{M}_{\text{source}}$, and a set of suspicious DNNs, denoted as $\{\mathcal{M}_1, \mathcal{M}_2, \cdots\}$. Some suspicious DNNs were unrelated to the source DNN $\mathcal{M}_{\text{source}}$, while others were obtained by sequentially applying the permutation attack, the scaling attack, and the fine-tuning attack[6] to $\mathcal{M}_{\text{source}}$. Then, for each pair of a source DNN $\mathcal{M}_{\text{source}}$ and a

---

[5]The magnitude and other specific settings of the noise $\epsilon$ will be introduced in Appendix B.1.

[6]Specifically, filters of all convolutional layers were first permuted, then scaled, and the DNN was finally fine-tuned on a new dataset. Please see Appendix C.2 for technical details.

Table 2: Classification accuracy (%) for determining whether a suspicious model originates from a source model.

| Methods | Fine-tuning set | CIFAR-10 | | | | | Imagenette | | | | |
|---|---|---|---|---|---|---|---|---|---|---|---|
| | Learning rate | 1 | $10^{-1}$ | $10^{-2}$ | $10^{-3}$ | $10^{-4}$ | 1 | $10^{-1}$ | $10^{-2}$ | $10^{-3}$ | $10^{-4}$ |
| RS ([Bansal et al., 2022]) | | 51.1 | 100.0 | 100.0 | 100.0 | 100.0 | 63.3 | 100.0 | 100.0 | 100.0 | 100.0 |
| MW ([Kim et al., 2023]) | | 50.0 | 54.4 | 95.0 | 100.0 | 100.0 | 51.1 | 50.0 | 100.0 | 100.0 | 100.0 |
| ICS ([Zeng et al., 2024]) | | 56.7 | 60.6 | 80.00 | 100.0 | 100.0 | 61.1 | 60.6 | 88.9 | 99.4 | 100.0 |
| CKA ([Zhang et al., 2024]) | | 55.0 | 57.2 | 80.0 | 100.0 | 100.0 | 55.6 | 58.3 | 100.0 | 100.0 | 100.0 |
| **Ours** | | 90.0 | 100.0 | 100.0 | 100.0 | 100.0 | 100.0 | 100.0 | 100.0 | 100.0 | 100.0 |

suspicious DNN $\mathcal{M}_i$, if the fingerprint score $\rho$ exceeded a threshold $t$ ($\rho > t$), then we considered these two DNNs to be of the same origin; otherwise not.

We compared our method with the following four competing fingerprinting methods or model source tracing methods. The first two methods were Random Smoothing (RS) [Bansal et al., 2022] and Marginal-based Watermarking (MW) [Kim et al., 2023], which used the classification results on the trigger set as the watermark. The inference scores of the two methods were defined as the trigger set accuracy of the suspicious DNN $\mathcal{M}_i$, denoted by $\rho_{\mathrm{RS}}$ and $\rho_{\mathrm{MW}}$, respectively. The third method was Cosine Similarity of Invariant Terms (ICS) [Zeng et al., 2024], which constructed an invariant term derived from the product of two-layer weights. The inference score $\rho_{\mathrm{ICS}}$ was defined as the cosine similarity of the invariant terms of $\mathcal{M}_{\mathrm{source}}$ and $\mathcal{M}_i$ (please see Appendix C.3 for technical details). The fourth method was Centered Kernel Alignment (CKA) [Zhang et al., 2024], which matched intermediate features as fingerprints via CKA similarity. The inference score $\rho_{\mathrm{CKA}}$ was defined as the CKA similarity of the intermediate features of $\mathcal{M}_{\mathrm{source}}$ and $\mathcal{M}_i$ (please see Appendix C.4 for technical details). Then, each method identified that a suspicious DNN $\mathcal{M}_i$ originated from a source DNN $\mathcal{M}_{\mathrm{source}}$ by checking whether the inference score exceeded a threshold $t$. The threshold $t$ was determined as the one that maximized the accuracy of model source tracing. For fair comparison, all these competing methods were evaluated on the same set of source DNNs and suspicious DNNs in the same way. The source DNNs with the ResNet-18 [He et al., 2016] architecture were trained on CIFAR-100 [Krizhevsky et al., 2009] with different random seeds. The fingerprint module was connected to the second convolutional layer of the second residual block.

Table 2 reports the accuracy of model source tracing achieved by our method under combined attacks. Our method achieved the highest accuracy of model source tracing across all datasets and different learning rates, especially when the learning rate was large. These results successfully verified the robustness of our method against the combined attacks of permutation, scaling, and fine-tuning.

**Verifying the robustness towards the individual attacks.** We conducted a set of experiments to separately verify the robustness of the fingerprint against the individual attacks, including the fine-tuning attack, the permutation attack, and the scaling attack. We applied each specific attack to a fingerprinted neural network and compared the fingerprint scores $\rho$ in Equation (14) before and after the attack (please see Appendix B.3 for results and more details). The results verified that the proposed fingerprint was robust against the fine-tuning attack, the permutation attack, and the scaling attack when these attacks were applied individually.

## 5   Acknowledgements

This work is partially supported by the National Science and Technology Major Project (2021ZD0111602), the National Nature Science Foundation of China (92370115, 62276165), and Shanghai Natural Science Foundation (24ZR1491700). It is also partially supported by Alibaba Group.

## 6   Conclusion

In this paper, we discover and theoretically prove that specific frequency components of a convolutional filter are invariant to model fine-tuning, and are equivariant to weight scaling and weight permutations. Therefore, we propose to use these frequency components as the network's fingerprint

to embed the ownership information so as to obtain a theoretically guaranteed robustness to the combined attacks of permutation, scaling, and fine-tuning. Additionally, to defend against the overwriting attack, we add an additional loss term during training to make sure that the network's performance drops significantly under the overwriting attack. Preliminary experiments have demonstrated the effectiveness of the proposed method.

This paper proves an ideal case where the fingerprint is exactly invariant to fine-tuning. Note that in real applications, the input feature contains low-frequency components, and the frequencies must be integers, making the fingerprint approximately invariant, rather than exactly unchanged. A more idealized setting can be achieved when the feature map size is large or the feature map size is divisible by the kernel size. Nonetheless, our experiments conducted on real networks demonstrate the robustness of the fingerprint even when the ideal conditions are not strictly satisfied.

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

# A  Proofs of our theoretical findings

We first introduce an important equation, which is widely used in the following proofs.

**Lemma A.1.** Given $N$ complex numbers, $e^{in\theta}$, $n = 0, 1, \ldots, N-1$, the sum of these $N$ complex numbers is given as follows.

$$\forall \theta \in \mathbb{R}, \qquad \sum_{n=0}^{N-1} e^{in\theta} = \frac{\sin(\frac{N\theta}{2})}{\sin(\frac{\theta}{2})} e^{i\frac{(N-1)\theta}{2}} \tag{16}$$

Specifically, when $N\theta = 2k\pi, k \in \mathbb{Z}, -N < k < N$, we have

$$\forall \theta \in \mathbb{R}, \quad \sum_{n=0}^{N-1} e^{in\theta} = \frac{\sin(\frac{N\theta}{2})}{\sin(\frac{\theta}{2})} e^{i\frac{(N-1)\theta}{2}} = N\delta_\theta; \quad \text{s.t. } N\theta = 2k\pi, k \in \mathbb{Z}, -N < k < N,$$

$$\text{where} \quad \delta_\theta = \begin{cases} 1, & \theta = 0 \\ 0, & \text{otherwise} \end{cases} \tag{17}$$

We prove Lemma A.1 as follows.

*Proof.* First, let us use the letter $S \in \mathbb{C}$ to denote the term of $\sum_{n=0}^{N-1} e^{in\theta}$.

$$S = \sum_{n=0}^{N-1} e^{in\theta}$$

Therefore, $e^{i\theta} S$ is formulated as follows.

$$e^{i\theta} S = \sum_{n=1}^{N} e^{in\theta} \in \mathbb{C}$$

Then, $S$ can be computed as $S = \frac{e^{i\theta} S - S}{e^{i\theta} - 1}$. Therefore, we have

$$
\begin{aligned}
S &= \frac{e^{i\theta} S - S}{e^{i\theta} - 1} \\
&= \frac{\sum_{n=1}^{N} e^{in\theta} - \sum_{n=0}^{N-1} e^{in\theta}}{e^{i\theta} - 1} \\
&= \frac{e^{iN\theta} - 1}{e^{i\theta} - 1} \\
&= \frac{e^{i\frac{N\theta}{2}} - e^{-i\frac{N\theta}{2}}}{e^{i\frac{\theta}{2}} - e^{-i\frac{\theta}{2}}} e^{i\frac{(N-1)\theta}{2}} \\
&= \frac{(e^{i\frac{N\theta}{2}} - e^{-i\frac{N\theta}{2}})/2i}{(e^{i\frac{\theta}{2}} - e^{-i\frac{\theta}{2}})/2i} e^{i\frac{(N-1)\theta}{2}} \\
&= \frac{\sin(\frac{N\theta}{2})}{\sin(\frac{\theta}{2})} e^{i\frac{(N-1)\theta}{2}}
\end{aligned}
$$

Therefore, we prove that $\sum_{n=0}^{N-1} e^{in\theta} = \frac{\sin(\frac{N\theta}{2})}{\sin(\frac{\theta}{2})} e^{i\frac{(N-1)\theta}{2}}$.

Then, we prove the special case that when $N\theta = 2k\pi, k \in \mathbb{Z}, -N < k < N$, $\sum_{n=0}^{N-1} e^{in\theta} = N\delta_\theta = \begin{cases} N, & \theta = 0 \\ 0, & \text{otherwise} \end{cases}$, as follows.

When $\theta = 0$, we have

$$\lim_{\theta \to 0} \sum_{n=0}^{N-1} e^{in\theta} = \lim_{\theta \to 0} \frac{\sin(\frac{N\theta}{2})}{\sin(\frac{\theta}{2})} e^{i\frac{(N-1)\theta}{2}}$$

$$= \lim_{\theta \to 0} \frac{\sin(\frac{N\theta}{2})}{\sin(\frac{\theta}{2})}$$

$$= N$$

When $\theta \neq 0$, and $N\theta = 2k\pi, k \in \mathbb{Z}, -N < k < N$, we have

$$\sum_{n=0}^{N-1} e^{in\theta} = \frac{\sin(\frac{N\theta}{2})}{\sin(\frac{\theta}{2})} e^{i\frac{(N-1)\theta}{2}}$$

$$= \frac{\sin(k\pi)}{\sin(\frac{k\pi}{N})} e^{i\frac{(N-1)k\pi}{N}}$$

$$= 0$$

$\square$

In the following proofs, the following two equations are widely used, which are derived based on Lemma A.1.

$$\sum_{m=0}^{M-1} \sum_{n=0}^{N-1} e^{-i(\frac{um}{M} + \frac{vn}{N})2\pi} = \sum_{m=0}^{M-1} e^{im(-\frac{u2\pi}{M})} \sum_{n=0}^{N-1} e^{in(-\frac{v2\pi}{N})}$$

$$= (M\delta_{-\frac{u2\pi}{M}})(N\delta_{-\frac{v2\pi}{N}}) \quad //\text{According to Equation (17)}$$

$$= \begin{cases} MN, & u = v = 0 \\ 0, & \text{otherwise} \end{cases}$$

To simplify the representation, **let $\delta_{uv}$ be the simplification of $\delta_{-\frac{u2\pi}{M}} \delta_{-\frac{v2\pi}{N}}$ in the following proofs.** Therefore, we have

$$\sum_{m=0}^{M-1} \sum_{n=0}^{N-1} e^{-i(\frac{um}{M} + \frac{vn}{N})2\pi} = MN\delta_{uv} = \begin{cases} MN, & u = v = 0 \\ 0, & \text{otherwise} \end{cases} \tag{18}$$

Similarly, we derive the second equation as follows.

$$\sum_{m=0}^{M-1} \sum_{n=0}^{N-1} e^{i(\frac{(u-u')m}{M} + \frac{(v-v')n}{N})2\pi} = \sum_{m=0}^{M-1} e^{im(\frac{(u-u')2\pi}{M})} \sum_{n=0}^{N-1} e^{in(\frac{(v-v')2\pi}{N})}$$

$$= MN\delta_{\frac{(u-u')2\pi}{M}} \delta_{\frac{(v-v')2\pi}{N}} \quad //\text{According to Equation (17)}$$

$$= MN\delta_{u-u'}\delta_{v-v'} \tag{19}$$

$$= \begin{cases} MN, & u' = u; v' = v \\ 0, & \text{otherwise} \end{cases}$$

## A.1 Proof of Theorem 3.2

In this section, we prove Theorem 3.2 in the main paper, as follows.

*Proof.* According to the DFT and the inverse DFT, we can obtain the mathematical relationship between $G_{uv}^{(c)}$ and $X_{mn}^{(c)}$, and the mathematical relationship between $Q_{uv}^{(c)}$ and $W_{ts}^{(c)}$, as follows.

$$\begin{cases} G_{uv}^{(c)} = \displaystyle\sum_{m=0}^{M-1}\sum_{n=0}^{N-1} X_{mn}^{(c)} e^{-i(\frac{um}{M}+\frac{vn}{N})2\pi} \\ X_{mn}^{(c)} = \dfrac{1}{MN}\displaystyle\sum_{u=0}^{M-1}\sum_{v=0}^{N-1} G_{uv}^{(c)} e^{i(\frac{um}{M}+\frac{vn}{N})2\pi} \end{cases} \begin{cases} Q_{uv}^{(c)} = \displaystyle\sum_{t=0}^{K-1}\sum_{s=0}^{K-1} W_{ts}^{(c)} e^{i(\frac{ut}{M}+\frac{vs}{N})2\pi} \\ W_{ts}^{(c)} = \dfrac{1}{MN}\displaystyle\sum_{u=0}^{M-1}\sum_{v=0}^{N-1} Q_{uv}^{(c)} e^{-i(\frac{ut}{M}+\frac{vs}{N})2\pi} \end{cases} \quad (20)$$

Based on Equation (20) and the derivation rule for complex numbers [Kreutz-Delgado, 2009], we can obtain the mathematical relationship between $\frac{\partial Loss}{\partial \overline{G}_{uv}^{(c)}}$ and $\frac{\partial Loss}{\partial \overline{X}_{mn}^{(c)}}$, and the mathematical relationship between $\frac{\partial Loss}{\partial \overline{Q}_{uv}^{(c)}}$ and $\frac{\partial Loss}{\partial \overline{W}_{ts}^{(c)}}$, as follows. Note that when we use gradient descent to optimize a real-valued loss function *Loss* with complex variables, people usually treat the real and imaginary values, $a \in \mathbb{C}$ and $b \in \mathbb{C}$, of a complex variable ($z = a + bi$) as two separate real-valued variables, and separately update these two real-valued variables. In this way, the exact optimization step of $z$ computed based on such a technology is equivalent to $\frac{\partial Loss}{\partial \overline{z}}$. Since $X_{mn}^{(c)}$ and $W_{cts}^{(l)[\mathrm{ker=d}]}$ are real numbers, $\frac{\partial Loss}{\partial \overline{X}_{mn}^{(c)}} = \frac{\partial Loss}{\partial X_{mn}^{(c)}}$ and $\frac{\partial Loss}{\partial \overline{W}_{ts}^{(c)}} = \frac{\partial Loss}{\partial W_{ts}^{(c)}}$.

$$\begin{cases} \dfrac{\partial Loss}{\partial \overline{G}_{uv}^{(c)}} = \dfrac{1}{MN}\displaystyle\sum_{m=0}^{M-1}\sum_{n=0}^{N-1} \dfrac{\partial Loss}{\partial \overline{X}_{mn}^{(c)}} e^{-i(\frac{um}{M}+\frac{vn}{N})2\pi} \\ \dfrac{\partial Loss}{\partial \overline{X}_{mn}^{(c)}} = \displaystyle\sum_{u=0}^{M-1}\sum_{v=0}^{N-1} \dfrac{\partial Loss}{\partial \overline{G}_{uv}^{(c)}} e^{i(\frac{um}{M}+\frac{vn}{N})2\pi} \end{cases} \begin{cases} \dfrac{\partial Loss}{\partial \overline{Q}_{uv}^{(c)}} = \dfrac{1}{MN}\displaystyle\sum_{t=0}^{K-1}\sum_{s=0}^{K-1} \dfrac{\partial Loss}{\partial \overline{W}_{ts}^{(c)}} e^{i(\frac{ut}{M}+\frac{vs}{N})2\pi} \\ \dfrac{\partial Loss}{\partial \overline{W}_{ts}^{(c)}} = \displaystyle\sum_{u=0}^{M-1}\sum_{v=0}^{N-1} \dfrac{\partial Loss}{\partial \overline{Q}_{uv}^{(c)}} e^{-i(\frac{ut}{M}+\frac{vs}{N})2\pi} \end{cases} \quad (21)$$

Let us conduct the convolution operation on the feature map $\mathbf{X} = [X^{(1)}, X^{(2)}, \cdots, X^{(C)}] \in \mathbb{R}^{C \times M \times N}$, and obtain the output feature map $Y \in \mathbb{R}^{M \times N}$ as follows.

$$Y_{mn} = b + \sum_{c=1}^{C}\sum_{t=0}^{K-1}\sum_{s=0}^{K-1} W_{ts}^{(c)} X_{m+t,n+s}^{(c)} \quad (22)$$

Based on Equation (20) and Equation (21), and the derivation rule for complex numbers [Kreutz-Delgado, 2009], the exact optimization step of $Q_{uv}^{(c)}$ in real implementations can be computed as

follows.

$$\frac{\partial Loss}{\partial \overline{Q}^{(c)}_{uv}}$$

$$= \frac{1}{MN} \sum_{t=0}^{K-1} \sum_{s=0}^{K-1} \frac{\partial Loss}{\partial \overline{W}^{(c)}_{ts}} e^{i(\frac{ut}{M} + \frac{vs}{N})2\pi} \quad //\text{Equation (21)}$$

$$= \frac{1}{MN} \sum_{t=0}^{K-1} \sum_{s=0}^{K-1} \left( \sum_{m=0}^{M-1} \sum_{n=0}^{N-1} \frac{\partial Loss}{\partial \overline{Y}_{mn}} \cdot \overline{X}^{(c)}_{m+t,n+s} \right) e^{i(\frac{ut}{M} + \frac{vs}{N})2\pi} \quad //\text{Equation (22)}$$

$$//\text{Equation (20)}$$

$$= \frac{1}{MN} \sum_{t=0}^{K-1} \sum_{s=0}^{K-1} \left( \sum_{m=0}^{M-1} \sum_{n=0}^{N-1} \frac{\partial Loss}{\partial \overline{Y}_{mn}} \cdot \frac{1}{MN} \sum_{u'=0}^{M-1} \sum_{v'=0}^{N-1} \overline{G}^{(c)}_{u'v'} e^{-i(\frac{u'(m+t)}{M} + \frac{v'(n+s)}{N})2\pi} \right) e^{i(\frac{ut}{M} + \frac{vs}{N})2\pi}$$

$$= \frac{1}{MN} \sum_{t=0}^{K-1} \sum_{s=0}^{K-1} \left( \sum_{u'=0}^{M-1} \sum_{v'=0}^{N-1} \overline{G}^{(c)}_{u'v'} e^{-i(\frac{u't}{M} + \frac{v's}{N})2\pi} \cdot \frac{1}{MN} \sum_{m=0}^{M-1} \sum_{n=0}^{N-1} \frac{\partial Loss}{\partial \overline{Y}_{mn}} e^{-i(\frac{u'm}{M} + \frac{v'n}{N})2\pi} \right) e^{i(\frac{ut}{M} + \frac{vs}{N})2\pi}$$

$$= \frac{1}{MN} \sum_{t=0}^{K-1} \sum_{s=0}^{K-1} \left( \sum_{u'=0}^{M-1} \sum_{v'=0}^{N-1} \overline{G}^{(c)}_{u'v'} \frac{\partial Loss}{\partial \overline{H}_{u'v'}} e^{-i(\frac{u't}{M} + \frac{v's}{N})2\pi} \right) e^{i(\frac{ut}{M} + \frac{vs}{N})2\pi} \quad //\text{Equation (21)}$$

$$= \frac{1}{MN} \sum_{t=0}^{K-1} \sum_{s=0}^{K-1} \sum_{u'=0}^{M-1} \sum_{v'=0}^{N-1} \overline{G}^{(c)}_{u'v'} \frac{\partial Loss}{\partial \overline{H}_{u'v'}} e^{i(\frac{(u-u')t}{M} + \frac{(v-v')s}{N})2\pi}$$

$$= \sum_{u'=0}^{M-1} \sum_{v'=0}^{N-1} \overline{G}^{(c)}_{u'v'} \frac{\partial Loss}{\partial \overline{H}_{u'v'}} \cdot \frac{1}{MN} \sum_{t=0}^{K-1} \sum_{s=0}^{K-1} e^{i(\frac{(u-u')t}{M} + \frac{(v-v')s}{N})2\pi}$$

$$// \text{ Let } A_{u'v'uv} = \sum_{t=0}^{K-1} \sum_{s=0}^{K-1} e^{i(\frac{(u-u')t}{M} + \frac{(v-v')s}{N})2\pi}$$

$$= \frac{1}{MN} \sum_{u'=0}^{M-1} \sum_{v'=0}^{N-1} \overline{G}^{(c)}_{u'v'} \frac{\partial Loss}{\partial \overline{H}_{u'v'}}$$

where $A_{u'v'uv}$ can be rewritten as follows.

$$A_{u'v'uv} = \sum_{t=0}^{K-1} \sum_{s=0}^{K-1} e^{i(\frac{(u-u')t}{M} + \frac{(v-v')s}{N})2\pi}$$

$$= \sum_{t=0}^{K-1} e^{i\frac{(u-u')2\pi}{M}t} \sum_{s=0}^{K-1} e^{i\frac{(v-v')2\pi}{N}s}$$

$$= \frac{\sin(\frac{K(u-u')\pi}{M})}{\sin(\frac{(u-u')\pi}{M})} \frac{\sin(\frac{K(v-v')\pi}{N})}{\sin(\frac{(v-v')\pi}{N})} \cdot e^{i(\frac{(K-1)(u-u')}{M} + \frac{(K-1)(v-v')}{N})\pi} \quad //\text{According to Equation (16)}$$

Based on the derived $\frac{\partial Loss}{\partial \overline{Q}^{(c)}_{uv}} \in \mathbb{C}$, we can further compute gradients $\frac{\partial Loss}{\partial (\overline{T}^{(l,uv)})^\top} \in \mathbb{C}^{D \times C}$ as follows.

$$\frac{\partial Loss}{\partial \overline{\mathcal{F}}_{\mathbf{W}}^{(uv)}} = \frac{1}{MN} \sum_{u'=0}^{M-1} \sum_{v'=0}^{N-1} A_{uvu'v'} \frac{\partial Loss}{\partial \overline{\mathcal{F}}_{Y}^{(u'v')}} \cdot \overline{\mathcal{F}}_{\mathbf{X}}^{(u'v')} \tag{23}$$

Let us use the gradient descent algorithm to update the convlutional weight $W^{(c)}_{ts}|_n$ of the $n$-th epoch, the updated frequency spectrum $W^{(c)}_{ts}|_{n+1}$ can be computed as follows.

$$\forall t, s, \quad W^{(c)}_{ts}|_{n+1} = W^{(c)}_{ts}|_n - \eta \cdot \frac{\partial Loss}{\partial \overline{W}^{(c)}_{ts}}$$

where $\eta$ is the learning rate. Then, the updated frequency spectrum $T^{(l,uv)}|_{n+1}$ computed based on Equation (21) is given as follows.

$$\Delta Q_{uv}^{(c)} = Q_{uv}^{(c)}|_{n+1} - Q_{uv}^{(c)}|_n$$

$$= \sum_{t=0}^{K-1}\sum_{s=0}^{K-1} W_{ts}^{(c)}|_{n+1} e^{i(\frac{ut}{M}+\frac{vs}{N})2\pi} - Q_{uv}^{(c)}|_n \quad //\text{Equation (20)}$$

$$= \sum_{t=0}^{K-1}\sum_{s=0}^{K-1} (W_{ts}^{(c)}|_n - \eta \cdot \frac{\partial Loss}{\partial \overline{W}_{ts}^{(c)}}) e^{i(\frac{ut}{M}+\frac{vs}{N})2\pi} - Q_{uv}^{(c)}|_n$$

$$= (\sum_{t=0}^{K-1}\sum_{s=0}^{K-1} W_{ts}^{(c)}|_n e^{i(\frac{ut}{M}+\frac{vs}{N})2\pi} - Q_{uv}^{(c)}|_n) - \eta \sum_{t=0}^{K-1}\sum_{s=0}^{K-1} \frac{\partial Loss}{\partial \overline{W}_{ts}^{(c)}} e^{i(\frac{ut}{M}+\frac{vs}{N})2\pi}$$

$$= -\eta \sum_{t=0}^{K-1}\sum_{s=0}^{K-1} \frac{\partial Loss}{\partial \overline{W}_{ts}^{(c)}} e^{i(\frac{ut}{M}+\frac{vs}{N})2\pi} \quad //\text{Equation (20)}$$

$$= -\eta MN \frac{\partial Loss}{\partial \overline{Q}_{uv}^{(c)}} \quad //\text{Equation (21)}$$

Therefore, we prove that any step on $W_{ts}^{(c)}$ equals to $MN$ step on $Q_{uv}^{(c)}$. In this way, the change of frequency components $\mathcal{F}_{\mathbf{W}}^{(uv)}$ can be computed as follows.

$$\Delta \mathcal{F}_{\mathbf{W}}^{(uv)} = -\eta \sum_{u'=0}^{M-1}\sum_{v'=0}^{N-1} A_{uvu'v'} \frac{\partial Loss}{\partial \overline{\mathcal{F}}_Y^{(u'v')}} \cdot \overline{\mathcal{F}}_{\mathbf{X}}^{(u'v')} \tag{24}$$

$\square$

## A.2 Proof of Corollary 3.3

In this section, we prove Corollary 3.3 in Section 3 of the main paper, as follows.

*Proof.* According to Theorem 3.2, the change of the frequency components at frequencies $(u, v) \in S$ can be further derived as follows.

$$\Delta \mathcal{F}_{\mathbf{W}}^{(uv)} = -\eta \sum_{u'=0}^{M-1}\sum_{v'=0}^{N-1} A_{uvu'v'} \frac{\partial Loss}{\partial \overline{\mathcal{F}}_Y^{(u'v')}} \cdot \overline{\mathcal{F}}_{\mathbf{X}}^{(u'v')}$$

$$= -\eta A_{uv00} \frac{\partial Loss}{\partial \overline{\mathcal{F}}_Y^{(00)}} \cdot \overline{\mathcal{F}}_{\mathbf{X}}^{(00)} \quad //\forall (u, v) \neq (0,0), \mathcal{F}_{\mathbf{X}}^{(uv)} = 0$$

$$= -\eta \frac{\partial Loss}{\partial \overline{\mathcal{F}}_Y^{(00)}} \cdot \overline{\mathcal{F}}_{\mathbf{X}}^{(00)} \cdot \frac{\sin(\frac{Ku\pi}{M})}{\sin(\frac{u\pi}{M})}\frac{\sin(\frac{Kv\pi}{N})}{\sin(\frac{v\pi}{N})} \cdot e^{i(\frac{(K-1)u}{M}+\frac{(K-1)v}{N})\pi}$$

$$= -\eta \frac{\partial Loss}{\partial \overline{\mathcal{F}}_Y^{(00)}} \cdot \overline{\mathcal{F}}_{\mathbf{X}}^{(00)} \cdot e^{i(\frac{(K-1)i\pi}{K}+\frac{(K-1)j\pi}{K})\pi} \cdot \frac{\sin(i\pi)}{\sin(i\pi/K)}\frac{\sin(j\pi)}{\sin(j\pi/K)}$$

$$//S = \{(u, v) \mid u = iM/K \text{ or } v = jN/K; \; i, j \in \{1, 2, \ldots, K-1\}\}$$

$$= 0 \quad // \sin(i\pi) = 0$$

Therefore, we have proved the frequency components $\mathcal{F}_{\mathbf{W}}^{(uv)}$ at the frequencies in the set $S$ keep invariant.

$\square$

## A.3 Proof of Theorem 3.5

In this section, we prove Theorem 3.5 in Section 3 of the main paper, as follows.

*Proof.* The frequency components $\mathcal{F}_{\mathbf{W}^*}^{(uv)}$ of the scaled filter $\mathbf{W}^* = a \cdot \mathbf{W}$ are computed as follows.

$$\mathcal{F}_{\mathbf{W}^*}^{(uv)} = [Q_{uv}^{*(1)}, Q_{uv}^{*(2)}, \ldots, Q_{uv}^{*(C)}]^\top$$

$$= [\sum_{t=0}^{K-1}\sum_{s=0}^{K-1} W_{ts}^{*(1)} e^{i(\frac{ut}{M}+\frac{vs}{N})2\pi}, \sum_{t=0}^{K-1}\sum_{s=0}^{K-1} W_{ts}^{*(2)} e^{i(\frac{ut}{M}+\frac{vs}{N})2\pi}, \cdots, \sum_{t=0}^{K-1}\sum_{s=0}^{K-1} W_{ts}^{*(C)} e^{i(\frac{ut}{M}+\frac{vs}{N})2\pi}]$$

$$= [\sum_{t=0}^{K-1}\sum_{s=0}^{K-1} a \cdot W_{ts}^{(1)} e^{i(\frac{ut}{M}+\frac{vs}{N})2\pi}, \sum_{t=0}^{K-1}\sum_{s=0}^{K-1} a \cdot W_{ts}^{(2)} e^{i(\frac{ut}{M}+\frac{vs}{N})2\pi}, \cdots, \sum_{t=0}^{K-1}\sum_{s=0}^{K-1} a \cdot W_{ts}^{(C)} e^{i(\frac{ut}{M}+\frac{vs}{N})2\pi}]^\top$$

$$= a \cdot [\sum_{t=0}^{K-1}\sum_{s=0}^{K-1} W_{ts}^{(1)} e^{i(\frac{ut}{M}+\frac{vs}{N})2\pi}, \sum_{t=0}^{K-1}\sum_{s=0}^{K-1} W_{ts}^{(2)} e^{i(\frac{ut}{M}+\frac{vs}{N})2\pi}, \cdots, \sum_{t=0}^{K-1}\sum_{s=0}^{K-1} W_{ts}^{(C)} e^{i(\frac{ut}{M}+\frac{vs}{N})2\pi}]^\top$$

$$= a \cdot [Q_{uv}^{(1)}, Q_{uv}^{(2)}, \ldots, Q_{uv}^{(C)}]^\top$$

$$= a \cdot \mathcal{F}_{\mathbf{W}}^{(uv)}$$

Thus, we have proved that the frequency components $\mathcal{F}_{\mathbf{W}^*}^{(uv)}$ of the scaled filter are equal to the scaled frequency components $a \cdot \mathcal{F}_{\mathbf{W}}^{(uv)}$ of the original filter.

$\square$

### A.4 Proof of Theorem 3.6

In this section, we prove Theorem 3.6 in Section 3 of the main paper, as follows.

*Proof.* The frequency components $[\mathcal{F}_{\mathbf{W}_{\pi(1)}}^{(uv)}, \cdots, \mathcal{F}_{\mathbf{W}_{\pi(D)}}^{(uv)}]$ of the permuted filters $[\mathbf{W}_{\pi(1)}, \mathbf{W}_{\pi(2)}, \cdots, \mathbf{W}_{\pi(D)}]$ are computed as follows.

$$\begin{aligned}\left[\mathcal{F}_{\mathbf{W}_{\pi(1)}}^{(uv)}, \cdots, \mathcal{F}_{\mathbf{W}_{\pi(D)}}^{(uv)}\right] &= \left[\mathcal{T}_{uv}(\mathbf{W}_{\pi(1)}), \mathcal{T}_{uv}(\mathbf{W}_{\pi(2)}), \cdots, \mathcal{T}_{uv}(\mathbf{W}_{\pi(D)}))\right] \quad //\text{Equation (4)}\\ &= \pi\left[\mathcal{T}_{uv}(\mathbf{W}_1), \mathcal{T}_{uv}(\mathbf{W}_2), \cdots, \mathcal{T}_{uv}(\mathbf{W}_D)\right] \\ &= \pi\left[\mathcal{F}_{\mathbf{W}_1}^{(uv)}, \cdots, \mathcal{F}_{\mathbf{W}_D}^{(uv)}\right]\end{aligned} \quad (25)$$

The frequency components $[\mathcal{F}_{\mathbf{W}_{\pi(1)}}^{(uv)}, \cdots, \mathcal{F}_{\mathbf{W}_{\pi(D)}}^{(uv)}]$ of the permuted filters $[\mathbf{W}_{\pi(1)}, \mathbf{W}_{\pi(2)}, \cdots, \mathbf{W}_{\pi(D)}]$ are equal to the permuated frequency components $\pi[\mathcal{F}_{\mathbf{W}_1}^{(uv)}, \cdots, \mathcal{F}_{\mathbf{W}_D}^{(uv)}]$.

$\square$

## B   More experimental results

### B.1   Ablation studies to evaluate the effectiveness of the newly added loss term

We conducted an ablation experiment to evaluate the effectiveness of the newly added loss term $\mathcal{L}_{\text{attack}}$, *i.e.*, examining whether the performance of the neural network was significantly hurt under the overwriting attack when the network was trained with the loss function $\mathcal{L}$ in Equation (15). We compared the classification accuracy of the network without the attack and the classification accuracy under the attack to analyze the performance decline of the network towards the overwriting attack.

We ran experiments of AlexNet [Krizhevsky et al., 2012] and ResNet-18 [He et al., 2016] on Caltech-101, Caltech-256 [Fei-Fei et al., 2006] (license unknown), CIFAR-10 and CIFAR-100 [Krizhevsky et al., 2009] (MIT License) for image classification tasks. For AlexNet, the fingerprint module containing $256$ convolutional filters was connected to the third convolutional layer. For ResNet-18, the fingerprint module containing $256$ convolutional filters was connected to the second convolutional layer of the second residual block. The scalar weight $\lambda$ was set to $5 \times 10^{-4}$. The noise $\epsilon$ added to the parameters in the fingerprint module was obtained by conducting the IDFT on a unit frequency component at a random frequency, and the $l_2$-norm of the noise $\epsilon$ was set to $0.5$ times the $l_2$-norm of

the weights. We trained the model using the SGD optimizer for 250 epochs, with a learning rate of $0.01$ for the first 100 epochs and $0.001$ for the remaining 150 epochs.

Table 3 shows the experimental results. We observe that if the network is trained with the loss function $\mathcal{L} = \mathcal{L}_{CE} + \mathcal{L}_{attack}$ in Equation (15), the classification accuracy significantly drops under the overwriting attack. The results indicate that the newly introduced loss term effectively defends the overwriting attack.

Table 3: Experimental results of the effectiveness of the newly added loss term $\mathcal{L}_{attack}$. Baseline denotes the test accuracy of a neural network normally trained without the fingerprint. With $\mathcal{L}_{CE} + \mathcal{L}_{attack}$ denotes the test accuracy of a fingerprinted network trained with the loss function $\mathcal{L}_{CE} + \mathcal{L}_{attack}$, and With $\mathcal{L}_{CE}$ denotes the test accuracy of a fingerprinted network trained without the added loss term $\mathcal{L}_{attack}$. The accuracy outside the bracket represents the accuracy of the network without the overwriting attack, and the accuracy inside the bracket represents the accuracy of the network under the overwriting attack.

| Dataset | Baseline (%) | | With $\mathcal{L}_{CE} + \mathcal{L}_{attack}$ (%) | | With $\mathcal{L}_{CE}$ (%) | |
|---|---|---|---|---|---|---|
| | AlexNet | ResNet-18 | AlexNet | ResNet-18 | AlexNet | ResNet-18 |
| CIFAR-10 | 91.03 | 94.83 | 90.28 (43.55) | 92.17 (72.26) | 91.12 (91.12) | 94.89 (94.89) |
| CIFAR-100 | 68.10 | 76.29 | 66.34 (36.93) | 75.49 (41.18) | 67.52 (67.52) | 76.53 (76.53) |
| Caltech-101 | 66.46 | 70.10 | 62.15 (32.53) | 67.14 (41.33) | 67.84 (67.84) | 69.87 (69.87) |
| Caltech-256 | 40.50 | 54.61 | 37.97 (15.37) | 50.33 (18.13) | 39.22 (39.22) | 53.86 (53.86) |

## B.2 Verifying the invariance of the frequency components

We conducted the an experiment to verify the invariance of the proposed fingerprint towards fine-tuning. Let us fine-tune a trained DNN with fingerprint module containing filters $[\mathbf{W}_1, \mathbf{W}_2, \cdots, \mathbf{W}_D]$, and obtain a fine-tuned DNN with filters $[\mathbf{W}'_1, \mathbf{W}'_2, \cdots, \mathbf{W}'_D]$. We computed the average the norm of the change of the frequency components $\mathbb{E}_d[\|\Delta \mathcal{F}^{(uv)}_{\mathbf{W}_d}\|]$ to measure the invariance of the proposed fingerprint, where $\Delta \mathcal{F}^{(uv)}_{\mathbf{W}_d} = \mathcal{F}^{(uv)}_{\mathbf{W}'_d} - \mathcal{F}^{(uv)}_{\mathbf{W}_d}$ denotes the change of the frequency components extracted from the $d$-th convolutional filter.

We trained AlexNet and ResNet-18 on CIFAR-100, and then fine-tuned them on ImageNette (Apache License 2.0), CIFAR-10 and Caltech-101 with the learning rate $0.01$ for 50 epochs. All other experimental settings remained the same as described in Section B.1. Figure 5 shows the results, verifying that the fingerprint is invariant to fine-tuning.

## B.3 Verifying the robustness towards fine-tuning attack, permutation attack, and scaling attack (individually applied)

**Verifying the robustness towards fine-tuning.** We conducted experiments to verify the robustness of the fingerprint against the fine-tuning attack. We applied the fine-tuning attack to a fingerprinted neural network and compared the fingerprint scores $\rho$ in Equation (14) before and after the attack.

We trained AlexNet and ResNet-18 on CIFAR-100, and then fine-tuned them on ImageNette, CIFAR-10 and Caltech-101 with the learning rate $0.01$ for 50 epochs. All other experimental settings remained the same as described in Section B.1. Table 4 shows the results. The results verified that the proposed fingerprint was robust against the fine-tuning attack.

**Verifying the robustness towards scaling attack.** We conducted experiments to verify the robustness of the proposed fingerprint towards scaling attack. Given a fingerprinted DNN, we scaled the parameters in the fingerprint module by a constant $a(a > 0)$, and then detected the fingerprint using the method introduced in Section 3.4. We used the fingerprint score DR to show the robustness of the fingerprint towards weight scaling. We trained AlexNet on CIFAR-10, CIFAR-100, Caltech-101 and Caltech-256. All other experiment settings remained the same as described in Section 3.5. Table 5 shows the Experimental results. All the fingerprint scores are $1.00$, showing that our method is highly robust to the weight scaling attack.

**Verifying the robustness towards permutation attack.** We conducted experiments to verify the robustness of the proposed fingerprint towards permutation attack. Given a fingerprinted DNN, we

Table 4: Experimental results of verifying the robustness towards fine-tuning. Baseline denotes the accuracy of a neural network normally trained without the fingerprint. Ours denotes the accuracy of a fingerprinted network. The score inside the bracket denotes the fingerprint score of the fine-tuned DNN. Accuracy outside the bracket denotes test accuracy on the dataset.

| Source | Target | Baseline | | Ours | |
|---|---|---|---|---|---|
| | | AlexNet | ResNet-18 | AlexNet | ResNet-18 |
| CIFAR-100 | ImageNette | 78.92% | 83.36% | 80.48% (1.00) | 86.23% (1.00) |
| | CIFAR-10 | 88.90% | 93.03% | 89.65% (1.00) | 94.12% (1.00) |
| | Caltech-101 | 70.02% | 76.80% | 72.11% (1.00) | 79.98% (1.00) |

Table 5: Experimental results of verifying the robustness towards scaling attack. The score outside the bracket denotes the fingerprint score without the weight scaling attack, and the score inside the bracket denotes the fingerprint score under the weight scaling attack.

| $a$ | CIFAR-10 | CIFAR-100 | Caltech-101 | Caltech-256 |
|---|---|---|---|---|
| 10 | 1.00 (1.00) | 1.00 (1.00) | 1.00 (1.00) | 1.00 (1.00) |
| 100 | 1.00 (1.00) | 1.00 (1.00) | 1.00 (1.00) | 1.00 (1.00) |

permuted the filters in the fingerprint module with a random permutation $\pi$, and then detected the fingerprint using the method introduced in Section 3.4. We used the fingerprint score DR to show the robustness of the fingerprint towards weight scaling. We trained AlexNet on CIFAR-10, CIFAR-100, Caltech-101 and Caltech-256. All other experiment settings remained the same as described in Section 3.5. Table 5 shows the Experimental results. All the fingerprint scores are $100\%$, showing that our method is highly robust to the weight permutation attack.

Table 6: Experimental results of verifying the robustness towards weight permutations. The score outside the bracket denotes the fingerprint score without the weight permutation attack, and the score inside the bracket denotes the fingerprint score under the weight permutation attack.

| $\pi$ | CIFAR-10 | CIFAR-100 | Caltech-101 | Caltech-256 |
|---|---|---|---|---|
| $\pi_1$ | 1.00 (1.00) | 1.00 (1.00) | 1.00 (1.00) | 1.00 (1.00) |
| $\pi_2$ | 1.00 (1.00) | 1.00 (1.00) | 1.00 (1.00) | 1.00 (1.00) |

### B.4 Compute resources

All DNNs can be trained within 6 hours on a single NVIDIA GeForce RTX 3090 GPU (with 24G GPU memory).

## C  Other details

### C.1 Details of the cosine similarity between two complex vectors

The cosine similarity $\cos(\mathbf{z}_1, \mathbf{z}_2)$ between two complex vectors $\mathbf{z}_1$ and $\mathbf{z}_2$ is defined as $\frac{\mathrm{Re}(\overline{\mathbf{z}_1} \cdot \mathbf{z}_2)}{\|\mathbf{z}_1\| \|\mathbf{z}_2\|}$, where $\overline{\mathbf{z}_1}$ denotes the conjugate of $\mathbf{z}_1$, $\|\mathbf{z}_1\|$ denotes the magnitude of $\mathbf{z}_1$, and $\mathrm{Re}(\cdot)$ represents the real part of a complex number. The cosine similarity, which ranges from $[-1, 1]$, measures the directional similarity between two complex vectors. When $\cos(\mathbf{z}_1, \mathbf{z}_2) = 1$, $\mathbf{z}_1$ and $\mathbf{z}_2$ have the same direction, while when $\cos(\mathbf{z}_1, \mathbf{z}_2) = -1$, $\mathbf{z}_1$ and $\mathbf{z}_2$ have opposite directions.

### C.2 Details of the combined attacks

We sequentially applying three types of attacks, including permutation, scaling, and fine-tuning on the source model.

**Permutation attack.**    In this step, for each ResNet block in the network, we randomly permute the output channels of the first convolutional layer. To preserve the functional integrity of the model, the

input channels of the subsequent convolutional layer within the same block are permuted using the same permutation order.

**Scaling attack.** Next, we apply a layer-wise scaling transformation. Specifically, the parameters of the first convolutional layer in each ResNet block are multiplied by a randomly sampled scaling factor from the range $[1/10, 10]$. To compensate for this modification and maintain the original output behavior, the parameters of the following convolutional layer are multiplied by the reciprocal of the same scaling factor.

**Fine-tuning attack.** Finally, we perform fine-tuning on the perturbed model using two datasets: CIFAR-10 and Imagenette. Each model is fine-tuned for 50 epochs under the same training settings to evaluate the effect of task adaptation on fingerprint invariance.

### C.3 Details of inference score for ICS

To calculate the inference score $\rho_{\text{ICS}}$ for ICS, we randomly select 100 images from the CIFAR-100 dataset as probing inputs. Each image is sequentially passed through the first and second convolutional layers within the second residual block of a ResNet-18 model. That is, the input first undergoes convolution with the filters of the first convolutional layer, and the output is then passed to the second convolutional layer. This fixed convolutional pipeline ensures that the final output is invariant to channel permutations and reciprocal scalings between these two layers.

For each input image, we obtain the final feature map output from the second convolutional layer, which is then element-wise multiplied with the original input image. The resulting product is flattened into a one-dimensional vector. This process is repeated for all 100 CIFAR-100 samples, and the resulting vectors are concatenated to form a single reference fingerprint vector for the model. To compute the inference score $\rho_{\text{ICS}}$ between two models, we repeat the above process for both models and calculate the cosine similarity between their respective fingerprint vectors.

### C.4 Details of inference score for CKA

To calculate the inference score $\rho_{\text{CKA}}$ for CKA, we randomly select $m = 100$ images from the CIFAR-100 dataset. For each image, we extract the feature map from the second convolutional layer in the second residual block of ResNet-18. These feature maps are flattened and stacked into two matrices $\mathbf{Z}, \mathbf{Z}' \in \mathbb{R}^{100 \times d}$, corresponding to the source and suspicious models.

We compute the RBF kernel matrices $K$ and $K'$ as:

$$K_{ij} = \exp\left(-\frac{|\mathbf{z}_i - \mathbf{z}j|^2}{2\sigma^2}\right), \quad K'ij = \exp\left(-\frac{|\mathbf{z}'_i - \mathbf{z}'_j|^2}{2\sigma^2}\right), \tag{26}$$

The CKA score is then computed as:

$$\rho_{\text{CKA}} = \text{CKA}(K, K') = \frac{\text{HISC}(K, K')}{\sqrt{\text{HISC}(K, K) \cdot \text{HISC}(K', K')}}, \tag{27}$$

where $\text{HISC}(K, K) = \frac{1}{(m-1)^2}\text{tr}(KHKH)$, $H = I - \frac{1}{m}\mathbf{1}\mathbf{1}^\top$.

## D  Societal impacts

**Potential positive societal impacts.** This work helps protect the ownership of deep learning models by adding a fingerprint that stays stable even after fine-tuning. This means if a model is stolen or changed, the owner can still prove it belongs to them. This can help stop model theft and make AI development more fair and safe. It can also make it easier to track where a model comes from, which is useful in real-world use, like in business or research.

**Potential negative societal impacts.** However, this technique could also cause some problems. For example, it might be used to secretly track how a model is used, which could harm user privacy or limit open research. It may also lead to arguments about who really owns a model, especially when people work together or use open-source models. In some cases, this method might make it harder for others to reuse or change a model freely.

