# OpenReview forum: "Towards the Resistance of Neural Network Fingerprinting to Fine-tuning"
_NeurIPS.cc/2025/Conference — NeurIPS 2025 poster_

### Official Review · Reviewer_yBqd · 2025-06-15

**Clarity:** 3
**Significance:** 3
**Originality:** 3
**Rating:** 4
**Confidence:** 3

**Summary:**

This paper proposes the first theoretical proof that low-frequency components of a convolutional filter are invariant to model fine-tuning, and are also equivariant to weight scaling and weight permutations. As such, this paper proposes to use these specific frequency components for fingerprinting. Besides providing theoretical proof, the authors also evaluated the proposed fingerprinting method against fine-tuning and combined weight attacks.

**Questions:**

Please consider addressing the weak points outlined above.

**Ethical Concerns:**

["NO or VERY MINOR ethics concerns only"]

**Limitations:**

The theoretical proof is provided for convolutional layers, which limits the applicability of the fingerprinting method to models with conv. architecture. The authors shall explain the limitations of the proposed fingerprinting method and whether the proof can be further generalized.

**Paper Formatting Concerns:**

I do not have concerns about paper formatting.

**Quality:**

3

**Strengths And Weaknesses:**

This paper has the following strengths:
+ The authors extended the previous works on fingerprinting robustness with theoretical proofs;
+ The authors performed an extensive evaluation of the proposed fingerprinting method against various combinations of attacks;
+ The authors compared the proposed fingerprinting method against existing counterparts and showed that the new method achieves a higher classification accuracy for detecting fine-tuned models.


This paper has the following weaknesses:
- The proposed fingerprinting method targets at convolutional layers, meaning that this method may not be applicable to popular language models.
- The experiment section does not include results about the accuracy of the model on the main tasks when using different fingerprinting methods.

---

> ### Author Rebuttal · Authors · 2025-07-31
>
> # Reply to Reviewer yBqd:
>
> Thank you for your efforts. We will answer all your questions. **If you have further concerns, please let us know, and we will update the response ASAP.**
>
> ## Q1. Concerns about the method’s applicability to non-convolutional models, such as language models.
>
> > Weakness 1. "The proposed fingerprinting method targets at convolutional layers, meaning that this method may not be applicable to popular language models."
> >
> > Limitation 1. "The theoretical proof is provided for convolutional layers, which limits the applicability of the fingerprinting method to models with conv. architecture. The authors shall explain the limitations of the proposed fingerprinting method and whether the proof can be further generalized."
> >
>
> Thanks a lot. We have followed your suggestions to **extend our methods on the Transformer-based architecture**. Because our fingerprint module is only connected in parallel to the neural network and takes part in the inference process, it can be widely applied to different DNNs without being affected by the network architecture. For a neural network with Transformer layers, we can connect our fingerprint module (in Section 3.3) in parallel to a Transformer layer.
>
> Therefore, we have conducted **new experiments** on Transformer-based architectures and larger-scale CNN-based networks to validate the generalizability and effectiveness of the proposed method.
>
> (1) We selected Transformer-based models, pretrained ViT-B/16  officially released in [1] and BERT-uncased [3]. For the ViT-B/16 model (a Transformer-based vision model), let us be given the input embedding tensor $X \in \mathbb{R}^{N \times D}$ of the sixth Transformer layer, where N is the number of tokens, and D is the hidden dimension. We reshape it into a 2D spatial representation $X’ \in \mathbb{R}^{D\times \sqrt{N}\times \sqrt{N}}$. This reshaped tensor is then passed through our fingerprint module. The output of the fingerprint module is subsequently reshaped to the corresponding dimensions and added as a residual to the output of the Transformer layer. The ViT-B/16 was fine-tuned on the Imagenette dataset.
>
> For the BERT-uncased model [3] (a language model), the fingerprint module was connected in parallel to the sixth Transformer layer and the model was fine-tuned on AG_NEWS [4].
>
> (2) The larger-scale CNN-based network is the ResNet-50 model [2] (a convolutional network). The fingerprint module was connected in parallel to the second convolutional layer of the second residual block of ResNet-50. This model was fine-tuned on CIFAR-100 dataset.
>
> All models were trained for 50 epochs with a batch size of 64. Other settings followed those in the main paper.
>
> We tested the fingerprint score (in Section 3.4) and the mean cosine similarity of fingerprinting frequency components before and after fine-tuning $E_{(u,v) \in S'}\ E_{d}\ [\cos(F^{(uv)}\_{Wd},\ F^{(uv)}\_{W'd})]$. We used $\tau = 0.98$ as the threshold for fingerprint score computation. The results are summarized in Table A and Table B below.
>
> Table A: The fingerprint scores of our method on Transformer-based architectures and larger-scale CNN-based networks.
>
> | Models \ Learning rates | 1e-2  | 1e-3  | 1e-4  |
> | :---------------------: | :---: | :---: | :---: |
> |      ViT-B/16 [1]       | 1.000 | 1.000 | 1.000 |
> |      ResNet-50 [2]      | 0.823 | 0.938 | 0.996 |
> |    BERT-uncased [3]     | 0.942 | 0.991 | 1.000 |
>
> Note that the fingerprint score represents the fraction of fingerprinting frequency components with cosine similarity exceeding the threshold $\tau = 0.98$, a fingerprint score above 0.8 is already highly favorable.
>
> Table B: The mean cosine similarity of fingerprinting frequency components before and after fine-tuning $E_{(u,v) \in S'}\ E_{d}\ [\cos(F^{(uv)}\_{Wd},\ F^{(uv)}\_{W'd})]$ of our method on Transformer-based architectures and larger-scale CNN-based networks.
>
> | Models \ Learning rates | 1e-2  | 1e-3  | 1e-4  |
> | :---------------------: | :---: | :---: | :---: |
> |      ViT-B/16 [1]       | 0.999 | 0.999 | 0.999 |
> |      ResNet-50 [2]      | 0.985 | 0.993 | 0.998 |
> |    BERT-uncased [3]     | 0.992 | 0.996 | 0.999 |
>
> We observed that both the fingerprint score and the average similarity remained high across different network architectures and learning rate settings, which together provided preliminary evidence that our method remained effective on Transformer-based architectures and larger-scale CNN-based networks.
>
> ***
>
> ## Q2. Missing comparison of main task accuracy across different fingerprinting methods.
>
> > Weakness 2. "The experiment section does not include results about the accuracy of the model on the main tasks when using different fingerprinting methods."
>
> Thanks a lot. We have followed your suggestion to conduct **a new experiment** to compare the impacts on the model’s performance of different fingerprint injection methods, and **our method caused less performance drop on the main task** compared to other mainstream fingerprinting methods.
>
> We trained the ResNet18 on CIFAR-100 and CIFAR-10 to inject the fingerprint. We tested the classification accuracy on the datasets. The compared methods were RS [1] and MW [2] mentioned in the main paper Section 4.
>
> Table C: The classification accuracy of the model.
>
> |          Methods           | CIFAR-10  | CIFAR-100 |
> | :------------------------: | :-------: | :-------: |
> |  **Without fingerprint**   | **94.8%** | **76.3%** |
> |  With fingerprint: RS [1]  |   84.2%   |   59.9%   |
> |  With fingerprint: MW [2]  |   87.8%   |   62.1%   |
> | **With fingerprint: Ours** | **92.2%** | **75.5%** |
>
> Table C shows the results. Compared to other mainstream fingerprinting methods, our method caused a smaller performance drop on the main task.
>
> ***
>
> ## References:
>
> [1] Dosovitskiy, Alexey, et al. "An image is worth 16x16 words: Transformers for image recognition at scale." *arXiv preprint arXiv:2010.11929* (2020).
>
> [2] He, Kaiming, et al. "Deep residual learning for image recognition." *Proceedings of the IEEE conference on computer vision and pattern recognition*. 2016.
>
> [3] Devlin, Jacob, et al. "Bert: Pre-training of deep bidirectional transformers for language understanding." *Proceedings of the 2019 conference of the North American chapter of the association for computational linguistics: human language technologies, volume 1 (long and short papers)*. 2019.
>
> [4] Zhang, Xiang, Junbo Zhao, and Yann LeCun. "Character-level convolutional networks for text classification." *Advances in neural information processing systems* 28 (2015).
>
> [5] Bansal, Arpit, et al. "Certified neural network watermarks with randomized smoothing." *International Conference on Machine Learning*. PMLR, 2022.
>
> [6] Kim, Byungjoo, et al. "Margin-based neural network watermarking." *International Conference on Machine Learning*. PMLR, 2023.

---

### Official Review · Reviewer_mgDF · 2025-06-29

**Clarity:** 2
**Significance:** 2
**Originality:** 3
**Rating:** 5
**Confidence:** 3

**Summary:**

This paper proposes a novel neural network fingerprinting method that embeds ownership information into deep neural networks by leveraging theoretically invariant frequency components of convolutional filters. The core idea is that when input features contain only low-frequency components, certain frequency elements of the filters remain stable during fine-tuning. The paper further demonstrates that these frequency components are also equivariant to weight scaling and permutation. Based on this, a fingerprint module is designed, along with a loss function to defend against overwriting attacks. Extensive experiments under fine-tuning, scaling, and permutation attacks show that the proposed method outperforms existing techniques.

**Questions:**

The paper theoretically demonstrates the invariance of specific frequency components in convolutional filters during fine-tuning and proposes a neural network fingerprinting mechanism based on this property. However, several questions remain:
1. Is the proposed method still effective in deeper convolutional neural networks (e.g., ResNet-50, ResNeXt), and can it maintain stability within more complex network architectures?

2. Can the method be extended to other types of neural architectures, or how applicable is it to non-CNN models such as Transformers or ViT?

3. The paper lacks critical information about the experimental setup, including the downstream tasks used for fine-tuning, detailed training configurations (e.g., optimizer, learning rate, number of epochs), and the number and construction methodology of the source and suspicious model sets.

4. The code is not publicly available, which compromises the reproducibility of the results and the practical applicability of the proposed method.

**Ethical Concerns:**

["NO or VERY MINOR ethics concerns only"]

**Final Justification:**

The authors have adequately addressed several of my concerns during the rebuttal. I have increased my score from 3 to 5, as I believe the main concerns were partially resolved and the paper can be valuable to the community with minor revisions.

**Limitations:**

The paper discusses the limitations of its work as well as the potential societal impacts it may bring.

**Quality:**

3

**Strengths And Weaknesses:**

Strengths:
1. This paper focuses on an important and emerging research direction: fingerprint embedding techniques for neural networks, which hold significant practical relevance and application value.
2. This paper proposes a theoretically grounded fingerprinting method and proves its robustness against fine-tuning, effectively addressing key limitations of prior approaches in terms of transferability and stability.
3. This paper thoroughly evaluates the proposed method under various attack scenarios and standard benchmarks, demonstrating its practicality and effectiveness.

Weaknesses:
1. The proposed method is only evaluated on CNN-based models, and its applicability and effectiveness on other neural architectures such as Transformers or Vision Transformers remain unclear.
2. This paper lacks detailed descriptions of the experimental setup.
3. The code and implementation details are not publicly released, which limits the reproducibility of the results and hinders the practical adoption of the method.

---

> ### Author Rebuttal · Authors · 2025-07-31
>
> # Reply to Reviewer mgDF:
>
> Thank you for your efforts. We will answer all your questions. **If you have further concerns, please let us know, and we will update the response ASAP.**
>
> ## Q1. Concerns about the generalizability and effectiveness of the proposed method on non-CNN architectures and deeper CNN architectures.
>
> > Weakness 1. "The proposed method is only evaluated on CNN-based models, and its applicability and effectiveness on other neural architectures such as Transformers or Vision Transformers remain unclear."
> >
> > Question 1. "Is the proposed method still effective in deeper convolutional neural networks (e.g., ResNet-50, ResNeXt), and can it maintain stability within more complex network architectures?"
> >
> > Question 2. "Can the method be extended to other types of neural architectures, or how applicable is it to non-CNN models such as Transformers or ViT?"
>
> Thanks a lot. We have followed your suggestions to **extend our methods on the Transformer-based architecture**. Because our fingerprint module is only connected in parallel to the neural network and takes part in the inference process, it can be widely applied to different DNNs without being affected by the network architecture. For a neural network with Transformer layers, we can connect our fingerprint module (in Section 3.3) in parallel to a Transformer layer.
>
> Therefore, we have conducted **new experiments** on Transformer-based architectures and larger-scale CNN-based networks to validate the generalizability and effectiveness of the proposed method.
>
> (1) We selected Transformer-based models, pretrained ViT-B/16  officially released in [1] and BERT-uncased [3]. For the ViT-B/16 model (a Transformer-based vision model), let us be given the input embedding tensor $X \in \mathbb{R}^{N \times D}$ of the sixth Transformer layer, where N is the number of tokens, and D is the hidden dimension. We reshape it into a 2D spatial representation $X’ \in \mathbb{R}^{D\times \sqrt{N}\times \sqrt{N}}$. This reshaped tensor is then passed through our fingerprint module. The output of the fingerprint module is subsequently reshaped to the corresponding dimensions and added as a residual to the output of the Transformer layer. The ViT-B/16 was fine-tuned on the Imagenette dataset.
>
> For the BERT-uncased model [3] (a language model), the fingerprint module was connected in parallel to the sixth Transformer layer and the model was fine-tuned on AG_NEWS [4].
>
> (2) The larger-scale CNN-based network is the ResNet-50 model [2] (a convolutional network). The fingerprint module was connected in parallel to the second convolutional layer of the second residual block of ResNet-50. This model was fine-tuned on CIFAR-100 dataset.
>
> All models were trained for 50 epochs with a batch size of 64. Other settings followed those in the main paper.
>
> We tested the fingerprint score (in Section 3.4) and the mean cosine similarity of fingerprinting frequency components before and after fine-tuning $E_{(u,v) \in S'}\ E_{d}\ [\cos(F^{(uv)}\_{Wd},\ F^{(uv)}\_{W'd})]$. We used $\tau = 0.98$ as the threshold for fingerprint score computation. The results are summarized in Table A and Table B below.
>
> Table A: The fingerprint scores of our method on Transformer-based architectures and larger-scale CNN-based networks.
>
> | Models \ Learning rates | 1e-2  | 1e-3  | 1e-4  |
> | :---------------------: | :---: | :---: | :---: |
> |      ViT-B/16 [1]       | 1.000 | 1.000 | 1.000 |
> |      ResNet-50 [2]      | 0.823 | 0.938 | 0.996 |
> |    BERT-uncased [3]     | 0.942 | 0.991 | 1.000 |
>
> Note that the fingerprint score represents the fraction of fingerprinting frequency components with cosine similarity exceeding the threshold $\tau = 0.98$, a fingerprint score above 0.8 is already highly favorable.
>
> Table B: The mean cosine similarity of fingerprinting frequency components before and after fine-tuning $E_{(u,v) \in S'}\ E_{d}\ [\cos(F^{(uv)}\_{Wd},\ F^{(uv)}\_{W'd})]$. of our method on Transformer-based architectures and larger-scale CNN-based networks.
>
> | Models \ Learning rates | 1e-2  | 1e-3  | 1e-4  |
> | :---------------------: | :---: | :---: | :---: |
> |      ViT-B/16 [1]       | 0.999 | 0.999 | 0.999 |
> |      ResNet-50 [2]      | 0.985 | 0.993 | 0.998 |
> |    BERT-uncased [3]     | 0.992 | 0.996 | 0.999 |
>
> We observed that both the fingerprint score and the average similarity remained high across different network architectures and learning rate settings, which together provided preliminary evidence that our method remained effective on Transformer-based architectures and larger-scale CNN-based networks.
>
> ***
>
> ## Q2. Lack of detailed descriptions of the experimental setup, including fine-tuning tasks, training settings, and model construction.
>
> > Weakness 2. "This paper lacks detailed descriptions of the experimental setup."
> >
> > Question 3. "The paper lacks critical information about the experimental setup, including the downstream tasks used for fine-tuning, detailed training configurations (e.g., optimizer, learning rate, number of epochs), and the number and construction methodology of the source and suspicious model sets."
>
> Thanks for your suggestion. Some of the experimental details, including the optimizer, learning rate, and number of training epochs are described in the Appendix (Lines 486–490, 503, and 520), and the hyperparameter settings of the fingerprint module are specified in the main paper (Line 220). Specifically, we used the SGD optimizer with a momentum of 0.9. During the fingerprint injection phase, models were trained for 250 epochs; during the fine-tuning phase, models were fine-tuned for 50 epochs, after which we observed that the training had converged.
>
> What’s more, unless otherwise stated, we used a default batch size of 64 and a learning rate of 0.01 to train the models with a fixed random seed of 0. All models were trained and fine-tuned for classification tasks. In order to construct source-suspicious model pairs, each source model was paired with every suspicious model in all possible combinations. Each source-suspicious model pair formed a sample for model tracing.
>
> If the paper is accepted, we will revise the manuscript to make these settings more explicitly presented in the main paper.
>
> ***
>
> ## Q3. Lack of public code and implementation details, limiting reproducibility and applicability.
>
> > Weakness 3. "The code and implementation details are not publicly released, which limits the reproducibility of the results and hinders the practical adoption of the method."
> >
> > Question 4. "The code is not publicly available, which compromises the reproducibility of the results and the practical applicability of the proposed method."
>
> Thanks for your suggestion. We have already open-sourced the code on GitHub. Due to the time limit of the rebuttal, the interface has not yet been fully polished, but all core functionalities have been released, and users can run the code successfully. We will continue to clean up the repository and provide clearer documentation in the final version.
>
> Due to the anonymity requirement of the review process, we are not allowed to put the GitHub link at this stage. We will release GitHub link for the code in the main paper if the paper is accepted. Thank you for your understanding.
>
> ***
>
> ## References:
>
> [1] Dosovitskiy, Alexey, et al. "An image is worth 16x16 words: Transformers for image recognition at scale." *arXiv preprint arXiv:2010.11929* (2020).
>
> [2] He, Kaiming, et al. "Deep residual learning for image recognition." *Proceedings of the IEEE conference on computer vision and pattern recognition*. 2016.
>
> [3] Devlin, Jacob, et al. "Bert: Pre-training of deep bidirectional transformers for language understanding." *Proceedings of the 2019 conference of the North American chapter of the association for computational linguistics: human language technologies, volume 1 (long and short papers)*. 2019.
>
> [4] Zhang, Xiang, Junbo Zhao, and Yann LeCun. "Character-level convolutional networks for text classification." *Advances in neural information processing systems* 28 (2015).

---

> > ### Comment · Reviewer_mgDF · 2025-08-03
> >
> > Thank you very much for your response. it has resolved my issue.
> > I’ll consider increasing my rating.

---

### Official Review · Reviewer_diWK · 2025-07-01

**Clarity:** 3
**Significance:** 3
**Originality:** 3
**Rating:** 5
**Confidence:** 2

**Summary:**

This paper proposes a robust DNN fingerprinting method designed to embed ownership information into models with theoretically guaranteed robustness against fine-tuning. They leverage specific frequency components of convolutional filters, which remain invariant during fine-tuning when the input features contain only low-frequency components. The authors propose a revised Fourier transform to extract these frequency components and theoretically prove their stability under various attacks. Additionally, they introduce an auxiliary class during the watermarking to defend against overwriting attacks via an auxiliary loss term. Experiments on datasets like CIFAR-10 and ImageNette demonstrate the method's effectiveness, outperforming existing techniques in robustness to fine-tuning.

**Questions:**

- Is it possible to extend your method to ViT-based architectures?
- How do you implement the overwriting attack in your experiments? Do you just add weight perturbations?
- Does the introduction of attack loss make your model more sensitive to input perturbation?
- Is there a more principal method to add the fingerprinting module? My concern is that it seems that the fingerprinting module somewhat depends on the model architecture. (Lines 479-481)
- What if the attacker first fine-tunes the fingerprinting layer and then uses the overwriting attack?

**Ethical Concerns:**

["NO or VERY MINOR ethics concerns only"]

**Final Justification:**

My primary concerns were addressed during the rebuttal, and I will keep my score unchanged.

**Limitations:**

yes

**Quality:**

3

**Strengths And Weaknesses:**

## Strengths
- This paper introduces a theoretically sound method with guaranteed robustness to fine-tuning, which largely strengthens the property protection of DNNs.
- Largely improved robustness compared to the baseline detection methods.
- Proofs are clear and complete.

## Weakness
- The proposed method requires an additional fingerprinting module and an auxiliary class to defend against the overwriting attacks. This weakens the threat model for the white-box property protection.

---

> ### Author Rebuttal · Authors · 2025-07-31
>
> # Reply to Reviewer diWK:
>
> Thank you for your efforts. We will answer all your questions. **If you have further concerns, please let us know, and we will update the response ASAP.**
>
> ## Q1. Concerns about white-box robustness
>
> > Weakness 1. "The proposed method requires an additional fingerprinting module and an auxiliary class to defend against the overwriting attacks. This weakens the threat model for the white-box property protection."
>
> Thank you for your response. We clarify the design of our method for defending against **white-box** and **black-box** attacks. Please correct us if we misunderstand anything.
>
> **For white-box attacks:** In our paper, the weight permutation attack, the weight scaling attack, and the overwriting attack are all typical white-box attacks, because these attacks require direct access to read and change the model’s parameters. We are not sure if the white-box attack you mentioned refers to these three types. If there is anything we misunderstood, please feel free to let us know. To this end, we have proven that our fingerprint is equivariant to permutation and scaling, thus robust to both. In addition, we also add an extra loss term to train the model to defend against the overwriting attack. Experiments in Section 3.5 and Section 4 showed that (1) the proposed fingerprint was robust against permutation attack and scaling attack, and (2) that overwriting the fingerprint significantly sacrificed the model’s performance.
>
> **For black-box attacks:** Our method mainly defends against the effect of standard fine-tuning on the fingerprint. We prove that certain frequency components used as the fingerprint remain stable during training in mathematics. Our experiments (see Section 4 for details) showed that the fingerprint persisted after fine-tuning, indicating that our method could effectively defend against the fine-tuning attack.
>
> ***
>
> ## Q2. Concerns about architectural dependency and extensibility
>
> > Question 1. "Is it possible to extend your method to ViT-based architectures?"
> >
> > Question 4. "Is there a more principal method to add the fingerprinting module? My concern is that it seems that the fingerprinting module somewhat depends on the model architecture. (Lines 479-481)"
>
> Thanks a lot. We have followed your suggestions to extend our method to Transformer-based architectures. Because our fingerprint module is only connected in parallel to the neural network and takes part in the inference process, it can be widely applied to different DNNs without being affected by the network architecture.
>
> Therefore, we have **conducted new experiments** on several models, including ViT-B/16 (a vision Transformer), BERT-uncased (a language model), and a larger CNN ResNet-50. However, due to the page limit, we are unable to include all the details and results in this rebuttal.
>
> Please refer to our **response to Reviewer yBqd**, particularly the **answer to Q1**, where we provided the experimental results and discussed this extension in more detail.
>
> ***
>
> ## Q3. How to implement the overwriting attack?
>
> > Question2. "How do you implement the overwriting attack in your experiments? Do you just add weight perturbations?"
>
> A good question. In fact, there are different ways to perform overwriting attacks. To better show the differences between them, we conducted **a new experiment** to test how the model performs under different types of overwriting. We applied the following three types of attacks:
>
> **(1) Replacement** [5]: This is also the most widely used overwriting attack method. This method replaced the convolutional weights in the fingerprint module with completely random values, and then fine-tuned the model on a new dataset to restore its performance.
>
> **(2) Noise on Spectrums**: This method conducted a revised Fourier transform on the convolutional weights in the fingerprint module, added Gaussian noise in the frequency domain (with an L2 norm of 0.5 times the original weights), and then applied an inverse transform to get the modified weights.
>
> **(3) Noise on Weights**: Another strategy directly added Gaussian noise to the convolutional weights of the fingerprint module (with an L2 norm of 0.5 times the original weights), which is quite similar to what you mentioned.
>
> We used ResNet-18 trained on CIFAR-100 dataset to inject the fingerprint. For fair comparison, we fine-tuned the models attacked by all types of overwriting, on the CIFAR-10 dataset. We then tested the fingerprint score (see Section 3.4) and classification accuracy on CIFAR-10. All other settings followed the main paper. Table B shows the results.
>
> Table A: The fingerprint score and the classification accuracy on CIFAR-10 after applying overwriting attacks to the models.
>
> | Attack types | Fingerprint score  | Acc. on CIFAR-10 |
> | :-: | :-: | :-: |
> |    **Without attack**     |     **1.00**    |**94.1%**   |
> |    With attack: Replacement      |     0.00    |82.4%   |
> |  With attack: Noise on Spectrums   | 0.08    |87.5%   |
> |   With attack: Noise on Weights    |     0.00    |88.0%   |
>
> In the experiments in the main paper, we used the second type of overwriting attack (Noise on Spectrums).
>
> Experimental results in the above table shows that all three types of attacks erased the fingerprint, but caused about 10% accuracy drop. This indicated that the overwriting attack could not erase the fingerprint without significantly hurting the performance.
>
> ***
>
> ## Q.4 Robustness against combined fine-tuning and overwriting attacks.
>
> > Question4. "What if the attacker first fine-tunes the fingerprinting layer and then uses the overwriting attack?"
>
> A good question. Let us answer your question from two perspectives.
>
> - First, during the fingerprint injection process, our method adds an extra adversarial loss $L_{attack}$ (see Equation (15)) to make the model’s performance drop when the overwriting attack is applied. This design ensures that the attacker cannot remove the fingerprint without hurting the model’s performance.
>
> - Second, we followed your suggestion and conducted **a new ablation experiment** to test the model’s performance in defending the “fine-tuning+overwriting” attack. The “fine-tuning+overwriting” attack was implemented as follows. Given a ResNet-18 model pretrained on CIFAR-100 for fingerprint injection, we applied two types of attack strategies. (1) The first strategy first fine-tuned the model on the CIFAR-10 dataset, and then overwrote the fingerprint module with completely new random parameters (not by adding noise). (2) The second reversed the order: the fingerprint module was first overwritten, and then the model was fine-tuned on the CIFAR-10 dataset.
>
> We applied this combined attack to two types of models: one was trained on CIFAR-100 dataset with both the cross-entropy loss $L_{CE}$ and the additional loss term $L_{attack}$, and the other one was trained only on CIFAR-100 dataset with the cross-entropy loss $L_{CE}$. Table B shows the results.
>
> Table B: The fingerprint score and classification accuracy on CIFAR-10 after applying the first strategy of the combined fine-tuning and overwriting attack to the models.
>
> |**Models**| **Fingerprint score before attack / after attack** | **Acc. on CIFAR-10 before attack / after attack** | Drop in acc. on CIFAR-10 after attack|
> | :-: | :-: | :-: | :-: |
> | ResNet-18 with fingerprint  model  learned without $L_{attack}$ |       1.00/0.00|94.2% / 93.7%|**0.5%**|
> | ResNet-18 with fingerprint model  learned with $L_{attack}$  |         1.00/0.00|94.1% / 25.3%|**68.8%**|
>
> Due to the page limit, we included the experimental results of the second combined attack strategy in the **response to Reviewer LxLU**, specifically in our **answer to Q1**.
>
> Under both attack strategies, we all observed that for the model trained with both $L_{CE}$ and $L_{attack}$, the fingerprint score dropped after the “fine-tuning + overwriting”  attack, but the model’s accuracy also decreased significantly. This shows that the extra loss term ensured that the attacker could not remove the fingerprint without hurting the performance.
>
> We will incorporate the above results into the final version of the paper.
>
> ***
>
> ## Q5. Does the loss $L_{attack}$ increase sensitivity to input perturbations?
>
> > Question3. "Does the introduction of attack loss make your model more sensitive to input perturbation?"
>
> A good question. We conducted an ablation study to check whether adding the additional loss term $L_{attack}$ made the model more sensitive to input perturbations. We perturbed the input samples by adding random Gaussian noise and measured the model’s final classification accuracy. The noise was scaled so that its L2 norm was 0.01, 0.05, 0.1, 0.5 times the L2 norm of the input sample. We ran the experiment on ResNet-18 with CIFAR-100 dataset, using the same experimental settings as in the main paper. The results are shown in the following table.
>
> Table C:
>
> |Models \ Noise scale| No noise | 0.01  | 0.05  |  0.1  |  0.5  |
> | :-: | :-: | :-: | :-: | :-: | :-: |
> |Normal ResNet-18 without the  fingerprint model|  76.3%   |76.2%|74.9%|67.5%|9.3%|
> | ResNet-18 with fingerprint  model  learned without $L_{attack}$ |  76.5%|76.4%|74.8%| 67.7% | 11.4% |
> | ResNet-18 with fingerprint model  learned with $L_{attack}$  |  75.5%   | 75.5% | 73.7% | 66.8% | 10.9% |
>
> The results showed that the added noise all hurt the classification accuracy of the three models, but the three models performed similarly when we added noises of different scales. This suggested that the loss $L_{attack}$ did not significantly affect the sensitivity to input perturbations.
>
> ***
>
> ## References:
>
> [1–4] Due to the page limit, please refer to the references in our **response to Reviewer yBqd** for the detailed citations [1–4], where the numbering remains consistent.
>
> [5] Wang et al. "Riga: Covert and robust white-box watermarking of deep neural networks." *Proceedings of the web conference 2021.*

---

### Official Review · Reviewer_LxLU · 2025-07-02

**Clarity:** 3
**Significance:** 3
**Originality:** 3
**Rating:** 4
**Confidence:** 3

**Summary:**

This paper presents a novel method to protect neural network models with convolutional layers by encoding fingerprint into the parameters of convolutional layers that corresponds to low-frequency components.

**Questions:**

Q1. I wonder if the authors can explain which types of attacks the method is designed to defend against and which ones are out of scope.

Q2. I would suggest to have a discussion: in which practical situations a drop in accuracy from adding the fingerprint would still be acceptable?

Q3. Can we have a discussion about why the L_{attack} term introduces this noticeable performance drop?

**Ethical Concerns:**

["NO or VERY MINOR ethics concerns only"]

**Final Justification:**

The author response has addressed many of my concerns. I have raised the score to 4.

**Limitations:**

Yes

**Quality:**

3

**Strengths And Weaknesses:**

Strong Points
----
S1. The paper introduces a theoretical analysis for some frequency components of the convolutional filter are stable during the training and fine-tuning.

S2. The experiments show that the fingerprint is hard to remove with regular fine-tuning or basic attacks.

Weak Points
----
W1. The paper does not test what happens if an attacker adds random noise to all the model's parameters and then fine-tunes the model. In real life, this is a common and strong way to try to erase a fingerprint. The paper mostly focuses on overwriting just the fingerprint module or on basic fine-tuning, so it's not clear if the method would still work against this stronger attack. It is also unclear that how the extra loss term L_{attack} term protect this kind of attack.

W2. In addition, when the authors train the model using the extra loss term extra loss term L_{attack}, the model's accuracy drops even before any attack happens. For example, accuracy on CIFAR-10 drops from 91.03% (baseline) to 90.28%, and on Caltech-101 it drops from 66.46% to 62.15%. This means the model gets worse just by adding the fingerprint, which could be a problem if high accuracy is needed.

---

> ### Author Rebuttal · Authors · 2025-07-29
>
> # Reply to Reviewer LxLU:
>
> Thank you for your efforts. We will answer all your questions. **If you have further concerns, please let us know, and we will update the response ASAP.**
>
> ## Q1. Robustness against combined overwriting and  fine-tuning attacks.
>
> > Weakness 1. "The paper does not test what happens if an attacker adds random noise to all the model's parameters and then fine-tunes the model. In real life, this is a common and strong way to try to erase a fingerprint. The paper mostly focuses on overwriting just the fingerprint module or on basic fine-tuning, so it's not clear if the method would still work against this stronger attack. It is also unclear that how the extra loss term $L_{attack}$ term protect this kind of attack."
>
> A good question. Let us answer your question from two perspectives.
>
> - First, during the fingerprint injection process, our method adds an extra adversarial loss $L_{attack}$ (see Equation (15)) to make the model’s performance drop when the overwriting attack is applied. This design ensures that the attacker cannot remove the fingerprint without hurting the model’s performance.
>
> - Second, we followed your suggestion and conducted **a new ablation experiment** to test the model’s performance to defend the “overwriting+fine-tuning” attack. The “overwriting+fine-tuning” attack was implemented as follows. Given a ResNet-18 model pretrained on CIFAR-100 for fingerprint injection, we first overwrote the fingerprint module with completely new random parameters (not by adding noise), and then fine-tuned the model on the CIFAR-10 dataset.
>
> We applied this combined attack to two types of models: one was trained on CIFAR-100 dataset with both the cross-entropy loss $L_{CE}$ and the additional loss term $L_{attack}$, and the other one was trained only on CIFAR-100 dataset with the cross-entropy loss $L_{CE}$. Table A shows the results.
>
> Table A: The fingerprint score and classification accuracy on CIFAR-10 after applying the combined overwriting and fine-tuning attacks to the models.
>
> |                          **Models**                          | **Fingerprint score before attack / after attack** | **Acc. on CIFAR-10 after attack** |
> | :----------------------------------------------------------: | :------------------------------------------------: | :-------------------------------: |
> | ResNet-18 with fingerprint  model  learned without $L_{attack}$ |                    1.00 / 0.00                     |             **93.3%**             |
> | ResNet-18 with fingerprint model  learned with $L_{attack}$  |                    1.00 / 0.00                     | 82.4% |
>
> We observed that for both the model trained with $L_{attack}$ and the model trained without $L_{attack}$, the fingerprint score was removed after the “overwriting + fine-tuning” attack. However, the classification accuracy of the model trained with $L_{attack}$ showed much lower accuracy than the model trained without $L_{attack}$. This indicated that the extra loss term ensured that the attacker could not remove the fingerprint without hurting the performance.
>
> - Besides, we also measured the classification accuracy on CIFAR-100 after only applying the overwriting attack, where the fingerprint was replaced with random new values (not by adding noise). Table B below shows the results.
>
> Table B: The classification accuracy on CIFAR-100 after applying the overwriting attack to the models.
>
> |                            Models                            | Acc. on CIFAR-100 before attack | Acc. on CIFAR-100 after overwriting attack | Drop in acc. after overwriting attack |
> | :----------------------------------------------------------: | :-----------------------------: | :----------------------------------------: | :-----------------------------------: |
> | ResNet-18 with fingerprint  model  learned without $L_{attack}$ |              76.5%              |                   76.2%                    |               **0.03%**               |
> | ResNet-18 with fingerprint model  learned with $L_{attack}$  |              75.5%              |                   36.5%                    |               **39.0%**               |
>
> We observed that for the model trained with the extra loss term $L_{\text{attack}}$, the accuracy dropped a lot (39.0%). But for the model trained without $L_{\text{attack}}$, the accuracy stayed almost the same (dropped by 0.3%). This indicated that even under a stronger overwriting setting, our method still prevented the attacker from removing the fingerprint without hurting the model’s performance.
>
> We will incorporate the above results into the final version of the paper.
>
> ***
>
> ## Q2. Accuracy drops due to the added fingerprinting loss.
>
> > Weakness 1. "In addition, when the authors train the model using the extra loss term extra loss term L_{attack}, the model's accuracy drops even before any attack happens. For example, accuracy on CIFAR-10 drops from 91.03% (baseline) to 90.28%, and on Caltech-101 it drops from 66.46% to 62.15%. This means the model gets worse just by adding the fingerprint, which could be a problem if high accuracy is needed."
> >
> > Question2. "I would suggest to have a discussion: in which practical situations a drop in accuracy from adding the fingerprint would still be acceptable?"
> >
> > Question3. "Can we have a discussion about why the L_attack term introduces this noticeable performance drop?"
>
> Thanks a lot. We add the extra loss term $L_{\text{attack}}$ to improve the model’s resistance to overwriting. This idea is somehow equivalent to changing the model’s adversarial robustness. As is well known, there is often a trade-off between adversarial robustness and classification accuracy. Changing the model’s adversarial robustness usually inevitably affects its classification performance.
>
> More crucially, we also conducted **a new experiment** to compare the impacts on the model’s performance of different fingerprint injection methods, and **our method caused less performance drop on the main task**, compared to other mainstream fingerprinting methods.
>
> We trained the ResNet18 on CIFAR-100 and CIFAR-10 to inject the fingerprint. We tested the classification accuracy on the datasets. The compared methods were RS [1] and MW [2] mentioned in the main paper Section 4.
>
> Table C: The classification accuracy of the model.
>
> |          Methods           | CIFAR-10  | CIFAR-100 |
> | :------------------------: | :-------: | :-------: |
> |  **Without fingerprint**   | **94.8%** | **76.3%** |
> |  With fingerprint: RS [1]  |   84.2%   |   59.9%   |
> |  With fingerprint: MW [2]  |   87.8%   |   62.1%   |
> | **With fingerprint: Ours** | **92.2%** | **75.5%** |
>
> Table C shows that compared to other mainstream fingerprinting methods, our method caused a smaller performance drop on the main task.
>
> ***
>
> ## Q3. Which types of attacks is the method designed to defend?
>
> > Question 1.  "I wonder if the authors can explain which types of attacks the method is designed to defend against and which ones are out of scope."
>
> Our method is designed to defend against fine-tuning, permutation, scaling, overwriting attacks, and combinations of these attack types. However, most existing non-backdoor watermarking methods [3,4,5,6] and ours operate on a different technical level from distillation attacks.
>
> The main contribution of our work is the proof of the first theoretically guaranteed fingerprint metric that is robust to fine-tuning. The technical differences between our method and the competing methods in Section 4 are summarized in Table D below. For comparisons with other mainstream methods, please refer to Table 1 in the main paper.
>
> Table D: Methods for model source tracing. “✓” indicates robustness, “✗” indicates lack of robustness, “–” means that the attack is not applicable, and “NTD” means that there is no target design to defend against the attack.
>
> | Methods |               Fine-tuning                | Permutation | Scaling | Overwriting | Pruning | Distilling |
> | :-----: | :--------------------------------------: | :---------: | :-----: | :---------: | :-----: | :--------: |
> | RS [1]  |       Enhanced via noise training        |      ✓      |    ✓    |      –      | No test |     ✓      |
> | MW [2]  | Enhanced via trigger confidence boosting |      ✓      |    ✓    |      –      | Tested  |     ✓      |
> | ICS [3] |              Empirical, NTD              |      ✓      |    ✓    |      ✗      | No test |     ✗      |
> | CKA[4]  |              Empirical, NTD              |      ✓      |    ✓    |      –      | Tested  |     ✗      |
> |  Ours   |         Theoretically guaranteed         |      ✓      |    ✓    |      ✓      | No test |     ✗      |
>
> ***
>
> ## References:
>
> [1] Bansal, Arpit, et al. "Certified neural network watermarks with randomized smoothing." *International Conference on Machine Learning*. PMLR, 2022.
>
> [2] Kim, Byungjoo, et al. "Margin-based neural network watermarking." *International Conference on Machine Learning*. PMLR, 2023.
>
> [3] Zeng, Boyi, et al. "Huref: Human-readable fingerprint for large language models." *Advances in Neural Information Processing Systems* 37 (2024): 126332-126362.
>
> [4] Zhang, Jie, et al. "Reef: Representation encoding fingerprints for large language models." *arXiv preprint arXiv:2410.14273* (2024).
>
> [5] Uchida, Yusuke, et al. "Embedding watermarks into deep neural networks." *Proceedings of the 2017 ACM on international conference on multimedia retrieval*. 2017.
>
> [6] Yang, Zhiguang, and Hanzhou Wu. "A fingerprint for large language models." *arXiv preprint arXiv:2407.01235* (2024).

---

> ### Comment · Reviewer_LxLU · 2025-08-01
>
> Thanks for the response. I will keep my score unchanged.

---

### Decision · Program_Chairs · 2025-09-17

**Decision:**

Accept (poster)

**Comment:**

This paper studies the problem of embedding ownership information into the weights of deep neural networks (DNNs). The authors provide a theoretical analysis showing that certain frequency components of convolutional filters remain stable during training and fine-tuning, which forms the foundation of the proposed approach. A revised Fourier transform is introduced to extract these frequency components, and theoretical guarantees are provided for their stability under various attacks. In addition, the method incorporates an auxiliary class during watermarking, with an auxiliary loss term to defend against overwriting attacks. Experiments on datasets such as CIFAR-10 and ImageNette demonstrate the effectiveness of the approach, showing improved robustness to fine-tuning compared to existing techniques.

All reviewers engaged in the discussion, acknowledged that most concerns were addressed, and recognized the theoretical analysis, noting that this paper makes a meaningful contribution to the area of robust model fingerprinting.